# Zatom-1:
# A Multimodal Flow Foundation Model
# for 3D Molecules and Materials

**Alex Morehead**[*,1], **Miruna Cretu**[*,2], **Antonia Panescu**[*,3], **Rishabh Anand**[*,3],
**Maurice Weiler**[*,4], **Tynan Perez**[*,4], **Samuel Blau**[1], **Steven Farrell**[1], **Wahid Bhimji**[1],
**Anubhav Jain**[1], **Hrushikesh Sahasrabuddhe**[1,5], **Pietro Liò**[2], **Tommi Jaakkola**[4],
**Rafael Gómez-Bombarelli**[4], **Rex Ying**[*,3], **N. Benjamin Erichson**[*,1,6], **Michael W. Mahoney**[*,1,5,6]

[1]Lawrence Berkeley National Laboratory*    [2]University of Cambridge    [3]Yale University
[4]MIT    [5]UC Berkeley    [6]ICSI

 github.com/Zatom-AI/zatom

## Abstract

General-purpose 3D chemical modeling encompasses molecules and materials, requiring both generative and predictive capabilities. However, most existing AI approaches are optimized for a single domain (molecules or materials) and a single task (generation or prediction), which limits representation sharing and transfer. We introduce Zatom-1, the first end-to-end, fully open-source foundation model that unifies generative and predictive learning of 3D molecules and materials. Zatom-1 is a Transformer trained with a multimodal flow matching objective that jointly models discrete atom types and continuous 3D geometries. This approach supports scalable pretraining with predictable gains as model capacity increases, while enabling fast and stable sampling. We use joint generative pretraining as a universal initialization for downstream multi-task prediction of properties, energies, and forces. Empirically, Zatom-1 matches or outperforms specialized baselines on both generative and predictive benchmarks, while reducing the generative inference time by more than an order of magnitude. Our experiments demonstrate positive predictive transfer between chemical domains from joint generative pretraining: modeling materials during pretraining improves molecular property prediction accuracy.

## 1 Introduction

Life, materials, and many modern functions, ranging from small molecule therapeutics to consumer electronic devices, are driven by chemistry. However, navigating the design space of chemical systems for high-impact downstream applications is notoriously time-consuming and error-prone (Blanco-Gonzalez et al., 2023; Venkatasubramanian, 2019; Wang et al., 2023). As such, developing our understanding of chemical systems with artificial intelligence (AI) methods represents a promising area of investment for deep learning research, given that these computational techniques have led to significant breakthroughs across related scientific fields in recent years (Jumper et al., 2021; Abramson et al., 2024).

Recently, deep learning has begun to unify the generation of new 3D molecules and materials (Joshi et al., 2025; Zhang et al., 2025a), and several specialized prediction models currently exist for advanced molecule and material property prediction (Liao & Smidt, 2023; Qu & Krishnapriyan, 2024). Nonetheless, no prior works offer an end-to-end, fully open-source model capable of both generative modeling and representation learning of 3D molecules and materials. To build such a unified model, several design choices must be addressed, including how to tokenize its inputs as well as what and how to learn using these tokens.

---

*denotes equal contribution

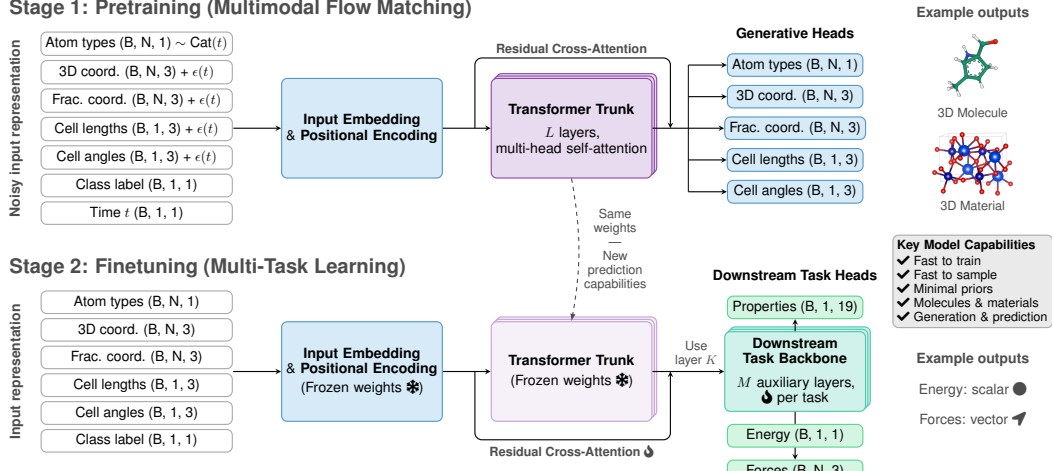

Figure 1: **Architecture of the ZATOM-1 foundation model for 3D molecules and materials.** A unified Transformer trunk processes (noisy) atomic, structural, and conditional inputs. The model features two training stages: (1) multimodal flow pretraining using the final trunk layer $L$'s representations to create new molecules or materials; and (2) multi-task finetuning of additional downstream task backbones and heads that use a specified trunk layer $K$'s representations to predict properties, energies, and forces.

Inspired by previous research on scientific foundation models (Subramanian et al., 2023; Abramson et al., 2024), we argue for the importance of tokenizing atomistic systems at the fundamental level of atomic granularity: 3D atom coordinates and element types. The next question is what to learn with these tokens. Previous works have shown that image and video diffusion models, employing standardized deep learning architectures, can learn rich embeddings for downstream prediction tasks in a self-supervised manner (Li et al., 2023; Vélez et al., 2025). In addition, scientific machine learning (SciML) is increasingly adopting standardized deep learning architectures to train highly scalable models (Abramson et al., 2024; Kreiman et al., 2025; Wadell et al., 2025). As such, it is clear that generative modeling, due to its self-supervised (i.e., label-free) nature, has the potential to serve as a powerful and flexible pretraining method for atomic foundation models.

In this spirit, we propose a question: *How much chemical knowledge can we represent with a 3D generative model?*

**ZATOM-1**, our answer to this question, is illustrated in Figure 1. It is a unified flow matching foundation model for 3D molecules and materials, based on the following ideas:

1. *Ambient all-atom generative modeling.* We represent 3D molecules and materials in $\mathbb{R}^3$ and perform generative flow matching directly within this Euclidean space. Importantly, with this approach, no pretrained autoencoders are necessary, e.g., for latent diffusion, delivering notable speed-ups in model training and inference.
2. *Representation learning using generative models.* We frame generative modeling of 3D molecules and materials as an ideal pretraining task for chemical representation learning (Briq et al., 2025) using modern, standardized Transformers (Vaswani et al., 2017).

**Contributions.** Based on these ideas, our main contributions in this work are as follows:

- We introduce ZATOM-1, a first-of-its-kind foundation model for diverse chemical tasks, including *de novo* generation of periodic materials and non-periodic 3D molecules, multi-task (and for materials zero-shot) property prediction, and energy and force modeling.
- ZATOM-1: (**1**) achieves competitive results for 3D materials generation on the LeMat-GenBench benchmark using the MP20 materials dataset; (**2**) exceeds state-of-the-art for 3D molecule generation on the QM9 and GEOM-Drugs molecular datasets and achieves state-of-the-art multi-task QM9 property prediction; (**3**) provides a 3x reduction in GPU

      training hours compared to a latent diffusion baseline; and (**4**) demonstrates broad, positive transfer learning through cross-domain generative modeling.

- ZATOM-1's architecture features a standardized Transformer with Query-Key Normalization (Wortsman et al., 2024), Flash Attention (Dao, 2024), and SwiGLU feed-forward networks (Shazeer, 2020), yielding predictably improving performance through parameter scaling. This also allows it to deliver a **12.5x** speed-up over a 500M parameter latent diffusion baseline, where ZATOM-1 (300M) can generate 10,000 samples in under 4 minutes with a single NVIDIA A100 GPU.

Overall, this work represents an advancement in chemical representation learning by introducing the first broadly applicable foundation model for 3D molecules and materials.

## 2 ZATOM-1

ZATOM-1's architecture is trained in two stages: (1) generative pretraining for 3D molecule and material generation; and (2) predictive finetuning for energy, force, and property prediction tasks. For generative pretraining, we use conditional flow matching, a scalable training technique for normalizing flows (Lipman et al., 2023). As a non-autoregressive generative modeling paradigm, conditional flow matching trains a neural network to approximate a time-conditioned velocity field required to transport samples from a source distribution to a target distribution using ordinary differential equation (ODE) integration. Importantly, time integration is only required during sampling, not model training, offering scalable training speeds as a result.

To efficiently model the multimodal nature of 3D atomic systems, we use multimodal flow matching over a (discrete) atom type modality and four (continuous) geometric modalities, yielding a joint distribution of these modalities for generative modeling. Previous works for 3D materials generation have attempted multimodal flow matching over similar modality structures (Miller et al., 2024). However, we argue that reliance on sparse graph neural network architectures, as well as hand-crafted generative priors, has hindered performance and training/inference speed for generative modeling in this domain (Krishnapriyan et al., 2021; Qu & Krishnapriyan, 2024). Instead, we adopt a standardized Transformer architecture and Gaussian multimodal flow matching for scalable joint modeling of 3D molecules and materials. We discuss related works in Appendix A.

### 2.1 PROBLEM FORMULATION

We aim to learn a joint generative model of 3D molecules and materials and to obtain rich neural embeddings of both data domains. Towards this end, we represent a 3D molecule or material with $N$ atoms using five unified modalities:

$$
\begin{aligned}
\text{Atom types } \boldsymbol{A} = \{a_i\}_{i=1}^N && \in \mathbb{Z}^{1 \times N}, \\
\text{3D coordinates } \boldsymbol{X} = \{x_i\}_{i=1}^N && \in \mathbb{R}^{3 \times N}, \\
\text{Fractional coordinates } \boldsymbol{F} = \{f_i\}_{i=1}^N && \in [0,1)^{3 \times N}, \\
\text{Unit cell/lattice lengths } \boldsymbol{L}_{\text{len}} = \{l_{\text{len}}\} && \in \mathbb{R}^{3 \times 1}, \\
\text{Unit cell/lattice angles } \boldsymbol{L}_{\text{ang}} = \{l_{\text{ang}}\} && \in \mathbb{R}^{3 \times 1}.
\end{aligned}
$$

Atom types $\boldsymbol{A}$ are a discrete modality defined over the set of integers $\mathbb{Z}$, while 3D coordinates $\boldsymbol{X}$ are a continuous modality of $\mathbb{R}^3$ expressed in Angstroms. Fractional coordinates $\boldsymbol{F} = \boldsymbol{L}^{-1}\boldsymbol{X}$ reside in the range $[0,1)$, and rotation-invariant lattice side lengths $\{l_{\text{len}}\} = \{a, b, c\} \in \mathbb{R}^{3 \times 1}$ and the three internal angles between them $\{l_{\text{ang}}\} = \{\alpha, \beta, \gamma\} \in [60°, 120°]^{3 \times 1}$ parametrize a parallelepiped $\boldsymbol{L} \in \mathbb{R}^{3 \times 3}$ (Grosse-Kunstleve et al., 2004) characterizing the repeating unit cell (Miller et al., 2024). For the sake of numerical stability, during training and sampling, lattice lengths are normalized by $\sqrt[3]{N}$, and lattice angles are converted from degrees to radians. Further, during training and sampling, for non-periodic molecules the lattice lengths, angles, and fractional coordinate inputs are masked to null values $\varnothing$. For periodic materials 3D coordinate inputs are masked to null values $\varnothing$ to enforce materials rotation invariance by default (with training losses masked accordingly).

**Algorithm 1** Pseudocode for TFT model $\mathcal{T}$

**Input:** Noisy 3D atoms $(\{a_i\}, \{x_i\}, \{f_i\}, l_{\text{len}}, l_{\text{ang}})$, class $c$, time $t$
**Output:** Denoised 3D atoms $(\{a_i'\}, \{x_i'\}, \{f_i'\}, l_{\text{len}}', l_{\text{ang}}')$,
       predictions $\{p^{\text{props'}}, p^{\text{energy'}}, p_i^{\text{forces'}}\}$

1: $\text{Lin}^{\text{B}}, \text{Lin}^{\text{NB}} = \text{Linear(bias=True), Linear(bias=False)}$
    # Mask inputs and project to $d_{\text{model}}$
2: $x_i, (f_i, l_{\text{len}}, l_{\text{ang}}) = x_i \cdot \mathbb{I}_{\text{molecule}}, (f_i, l_{\text{len}}, l_{\text{ang}}) \cdot \mathbb{I}_{\text{material}}$
3: $h_i = \text{Embed}(a_i) + \text{Embed}(c) + \text{Embed}(t) + \quad h_i \in \mathbb{R}^{d_{\text{model}}}$
    $\sum_{v \in \{x_i, f_i, l_{\text{len}}, l_{\text{ang}}\}} \text{Lin}_v^{\text{NB}}(v)$
    # Apply encoder trunk ($L$ layer transformer)
4: $\{z_i, z_i^{\text{K}}\} = \text{TransformerEncoder}^L(\{h_i\})$
    # Apply residual cross-attention to input embeddings
5: $\{h_i^a, h_i^x, h_i^f, h_i^{l_{\text{len}}}, h_i^{l_{\text{ang}}}\} = \text{TransformerDecoderBlock}(\{z_i, h_i\})$
    # Denoise inputs
6: $a_i' = \arg\max(\text{Lin}^{\text{B}}(\text{SiLU}(\text{Lin}^{\text{B}}(h_i^a))))$      $a_i' \in \mathbb{Z}$
7: $x_i' = \text{Lin}^{\text{NB}}(\text{LayerNorm}(h_i^x))$      $x_i' \in \mathbb{R}^3$
8: $f_i' = \text{Lin}^{\text{NB}}(\text{LayerNorm}(h_i^f))$      $f_i' \in \mathbb{R}^3$
9: $l_{\text{len}}' = \text{Lin}^{\text{NB}}\left(\text{LayerNorm}\left(\frac{1}{N}\sum_{i=1}^N h_i^{l_{\text{len}}}\right)\right)$   $l' \in \mathbb{R}^3$
10: $l_{\text{ang}}' = \text{Lin}^{\text{NB}}\left(\text{LayerNorm}\left(\frac{1}{N}\sum_{i=1}^N h_i^{l_{\text{ang}}}\right)\right)$   $l' \in \mathbb{R}^3$
    # Predict outputs (cross-attention + $M$ layer transformer)
11: $\{h_i^{\text{props}}, h_i^{\text{energy}}, h_i^{\text{forces}}\} = \text{TransformerDecoderBlock}_y(\{z_i^{\text{K}}, h_i\})$,
    for $y \in \{\text{props, energy, forces}\}$
12: Update $\{h_i^y\} \leftarrow \text{TransformerEncoder}_y^M(\{h_i^y\})$
13: $p^{\text{props'}} = \text{Lin}^{\text{B}}\left(\frac{1}{N}\sum_{i=1}^N h_i^{\text{props}}\right)$      $p \in \mathbb{R}^{19}$
14: $p^{\text{energy'}} = \text{Lin}^{\text{B}}\left(\frac{1}{N}\sum_{i=1}^N h_i^{\text{energy}}\right)$      $p \in \mathbb{R}^1$
15: $p_i^{\text{forces'}} = \text{Lin}^{\text{NB}}\left(h_i^{\text{forces}}\right)$      $p \in \mathbb{R}^3$

---

**Algorithm 2** Pseudocode for multimodal sampling

1: **Input:** Denoising model $\mathcal{T}$, number of atoms $N$, sampling steps $\{t_i\}_{i=0}^S$
2: **Output:** Sampled 3D atoms $(\mathbf{a}, \mathbf{x}, \mathbf{f}, \mathbf{l}_{\text{len}}, \mathbf{l}_{\text{ang}})$

3: **def** DISCRETEFLOWSTEP$(\mathbf{a}_t, \mathbf{p}_{t,a}', t, \Delta t)$:
    # NOTE: All indexed ops are vectorized
4:   $\mathbf{r}_t(i, \cdot) \leftarrow \frac{\Delta t}{1-t} \mathbf{p}_{t,a}'(i)$ (Campbell et al., 2024)
5:   $\mathbf{r}_t(i, \mathbf{a}_t(i)) \leftarrow -\sum_{j \neq \mathbf{a}_t(i)} \mathbf{r}_t(i, j)$
6:   $\mathbf{p}_{t+\Delta t}(i, j) \leftarrow \mathbb{1}_{\mathbf{a}_t(i)=j} + \mathbf{r}_t(i, j)$
7:   $\mathbf{a}_t(i) \leftarrow \text{Categorical}(\mathbf{p}_{t+\Delta t}(i, \cdot))$
8:   **return** $\mathbf{a}_t$

9: **def** EUCLIDEANSTEP$(\mathbf{z}_t, \mathbf{z}_t', t, \Delta t, g(\cdot), \gamma_g)$:
10:   $\mathbf{v}_t \leftarrow \frac{1}{1-t}(\mathbf{z}_t' - \mathbf{z}_t)$ (Geffner et al., 2025)
11:   $\mathbf{s}_t \leftarrow g(t) \frac{t\mathbf{v}_t - \mathbf{z}_t}{1-t}$
12:   $d\mathbf{W}_t \leftarrow \sqrt{2\,\gamma_g\,g(t)}\,\mathcal{N}(0, I)$
13:   $\mathbf{z}_t \leftarrow (\mathbf{v}_t + \mathbf{s}_t + d\mathbf{W}_t)\Delta t$
14:   **return** $\mathbf{z}_t$

    # Initialize all modalities with random noise
15: $\mathbf{a}_t \leftarrow \text{Categorical}(\delta(\frac{1}{\# \text{ atom types}}))$
16: Initialize $\mathbf{z}_t \leftarrow \mathcal{N}(0, I)$ for each continuous modality $\mathbf{z} \in \{\mathbf{x}, \mathbf{f}, \mathbf{l}_{\text{len}}, \mathbf{l}_{\text{ang}}\}$
    # Iteratively denoise over sampling steps
17: **for** $i = 1$ to $S$ **do**
18:   $\Delta t \leftarrow t_i - t_{i-1}$
    # Predict denoised endpoints using the model $\mathcal{T}$
19:   $(\mathbf{p}_{t,a}', \mathbf{x}_t', \mathbf{f}_t', \mathbf{l}_{t,\text{len}}', \mathbf{l}_{t,\text{ang}}') \leftarrow$
    $\mathcal{T}(\mathbf{a}_t, \mathbf{x}_t, \mathbf{f}_t, \mathbf{l}_{t,\text{len}}, \mathbf{l}_{t,\text{ang}}, t_i)$
    # Perform one sampling step for each modality
20:   $\mathbf{a}_t \leftarrow \text{DISCRETEFLOWSTEP}(\mathbf{a}_t, \mathbf{p}_{t,a}', t_i, \Delta t)$
21:   Update $\mathbf{z}_t \leftarrow \text{EUCLIDEANSTEP}(\mathbf{z}_t, \mathbf{z}_t', t_i, \Delta t)$ for each $\mathbf{z}$
22: **end for**
    # Return final denoised sample
23: **return** $(\mathbf{a}, \mathbf{x}, \mathbf{f}, \mathbf{l}_{\text{len}}, \mathbf{l}_{\text{ang}})$

## 2.2 NETWORK ARCHITECTURE

We use a standard Transformer architecture, titled the **Trunk-based Flow Transformer (TFT)**, to learn a shared latent representation of each molecular and material modality. For a given 3D atomic system $(\boldsymbol{A}, \boldsymbol{X}, \boldsymbol{F}, \boldsymbol{L}_{\text{len}}, \boldsymbol{L}_{\text{ang}})$ with a periodic (material) or non-periodic (molecule) class label $c$ and (optional) additive noise, a TFT model $\mathcal{T}$ with $L$ Transformer trunk layers learns latent representations $\boldsymbol{Z} = \{z_i\}_{i=1}^N \in \mathbb{R}^{d \times N}$ from each molecular and material modality to predict denoised modalities $\boldsymbol{S} = (\boldsymbol{A}', \boldsymbol{X}', \boldsymbol{F}', \boldsymbol{L}_{\text{len}}', \boldsymbol{L}_{\text{ang}}')$ and auxiliary properties $\boldsymbol{P} = (\boldsymbol{P}^{\text{props'}}, \boldsymbol{P}^{\text{energy'}}, \boldsymbol{P}^{\text{forces'}}) = (p^{\text{props'}} \in \mathbb{R}^{19}, p^{\text{energy'}} \in \mathbb{R}^1, \{p_i^{\text{forces'}}\}_{i=1}^N \in \mathbb{R}^{3 \times N})$:

$$\boldsymbol{S}, \boldsymbol{P} = \mathcal{T}(\boldsymbol{A}, \boldsymbol{X}, \boldsymbol{F}, \boldsymbol{L}_{\text{len}}, \boldsymbol{L}_{\text{ang}}, c), \tag{1}$$

where class label dropout with a 10% drop probability is applied during training to support (optional) classifier-free guidance during inference.

Algorithm 1 provides pseudocode for the forward pass of a TFT model with $L$ trunk layers, where layer $K \leq L$ is used for auxiliary task representations. Using random data augmentations during training, the model learns the rotational and periodic symmetries of molecules and materials, respectively (Abramson et al., 2024; Joshi et al., 2025). Additionally, inspired by the Platonic Transformer (Islam et al., 2025), we experiment with an O(3)-equivariant version of ZATOM-1 (PLATOM-1) featuring a TFT model variant titled the **Trunk-based Flow Platoformer (TFP)**, which we describe in more detail in Appendix B.

## 2.3 STAGE 1: MULTIMODAL FLOW PRETRAINING

At a high level, given a pair of data samples $(\mathbf{x}_0, \mathbf{x}_1)$ from their respective source and target distributions, flow matching (FM) offers a scalable technique for training generative models. Namely, using FM, one can train a neural network $v_\theta(\mathbf{x}_t, t)$ to sample data from the target distribution by matching the expected velocities transporting samples from the (often simpler) source to the target distribution:

$$\mathcal{L}_{\text{FM}} = \mathbb{E}_{t, (\mathbf{x}_0, \mathbf{x}_1)}[\|v_\theta(\mathbf{x}_t, t) - u_t\|_2^2], \tag{2}$$

where $\mathbf{x}_t = (1 - t)\mathbf{x}_0 + t\mathbf{x}_1$ and $u_t = \mathbf{x}_1 - \mathbf{x}_0$.

In this work, to optimize continuous modalities, we use Euclidean diffusion/conditional flow matching (CFM) (Sohl-Dickstein et al., 2015; Song & Ermon, 2019; Ho et al., 2020; Lipman et al., 2023; Albergo & Vanden-Eijnden, 2023; Tong et al., 2024), leveraging the equivalence between these two methods (Gao et al., 2024). For optimizing atom types, we employ Discrete CFM, which is parameterized within the framework of Discrete Flow Models (Campbell et al., 2024).

Specifically, consider a molecule or material comprised of $N$ atoms, characterized by zero-centered ground-truth 3D coordinates $\boldsymbol{X}$, fractional coordinates $\boldsymbol{F}$, lattice lengths $\boldsymbol{L}_{\text{len}}$, angles $\boldsymbol{L}_{\text{ang}}$, and atom types $\boldsymbol{A}$. The continuous 3D coordinates and fractional coordinates, along with lattice lengths and angles, undergo partial noise perturbation represented as $\boldsymbol{X}_t = t \cdot \boldsymbol{X} + (1-t) \cdot \epsilon$, where the noise $\epsilon$ is drawn from a Gaussian distribution $\mathcal{N}(0, I)$. For the discrete modality, the noisy atom types $\boldsymbol{A}_t$ are derived by interpolating between atom type probabilities and sampling from a categorical distribution, expressed as $\boldsymbol{A}_t \sim \text{Cat}(t \cdot \delta(\boldsymbol{A}) + (1-t) \cdot \delta(\frac{1}{\#\text{ atom types}}))$, where $\delta(\cdot)$ generates a one-hot encoding (Campbell et al., 2024). During the training phase, the model receives $\left(\boldsymbol{X}_t, \boldsymbol{F}_t, \boldsymbol{L}_{t_{\text{len}}}, \boldsymbol{L}_{t_{\text{ang}}}, \boldsymbol{A}_t\right)$ as input and learns to predict the terminal points of the trajectory.

**Flow matching losses.** The objective function for each continuous modality is given by

$$L_{\text{metric}}(\boldsymbol{X}) = \mathbb{E}_{\epsilon, t} \left[ \frac{1}{N} \|\boldsymbol{X}' - \boldsymbol{X}\|_2^2 \right]. \tag{3}$$

For the discrete atom types, the objective is defined using cross-entropy:

$$L_{\text{discrete}}(\boldsymbol{A}) = \mathbb{E}_t \left[ -\sum_i a_i \log(a_i') \right]. \tag{4}$$

These objectives are consolidated into a multi-objective framework, employing a weighting factor $\lambda_{\text{discrete}} \in [0, 1]$, expressed as

$$\begin{aligned} L_{\text{total}} = {}& L_{\text{metric}}(\boldsymbol{X}) + L_{\text{metric}}(\boldsymbol{F}) \\ & + L_{\text{metric}}(\boldsymbol{L}_{\text{len}}) + L_{\text{metric}}(\boldsymbol{L}_{\text{ang}}) \\ & + \lambda_{\text{discrete}} \cdot L_{\text{discrete}}(\boldsymbol{A}). \end{aligned} \tag{5}$$

During training, we set $\lambda_{\text{discrete}} = 0.1$ and sample from $t \sim \text{Beta}(\alpha_t, 1)$, where $\alpha_t = 1.8$, in line with prior studies (Geffner et al., 2025; Vonessen et al., 2025). Further, to learn data symmetries and accelerate convergence during training, we randomly rotate and (for periodic materials) translate multiple copies of each input example, which are noised to different levels according to batch-wise $t$ (Abramson et al., 2024). As $t \to 1$, the model resembles an identity function, since we adopt an endpoint formulation for flow matching due to its empirical performance for scientific applications (Stark et al., 2024). To enable the model to effectively learn accurate atom placements even when $t \to 1$ causes the losses to approach zero, based on the sampled time $t$, we scale the loss with $\beta(t) \cdot L_{\text{total}}(\boldsymbol{X}_t, \boldsymbol{F}_t, \boldsymbol{L}_{t_{\text{len}}}, \boldsymbol{L}_{t_{\text{ang}}}, \boldsymbol{A}_t)$ via:

$$\beta(t) = \min\left\{100, \ \frac{1}{(1-t)^2}\right\}. \tag{6}$$

**Multimodal sampling.** Algorithm 2 describes how we perform unconditional sampling of each continuous and discrete modality to generate 3D molecules and materials. Concisely, we sample random atom types or Gaussian noise for discrete and continuous modalities, respectively, and iteratively denoise each using a series of Euler steps operating on the multimodal predictions of the TFT model $\mathcal{T}$. Notably, other ODE solvers can be used, though we find that multimodal Euler steps provide strong empirical performance without considerable hyperparameter tuning.

## 2.4 STAGE 2: FINETUNING

As illustrated in Figure 1, we treat conditional flow matching of the modalities $(\boldsymbol{A}, \boldsymbol{X}, \boldsymbol{F}, \boldsymbol{L}_{\text{len}}, \boldsymbol{L}_{\text{ang}})$ as a self-supervised pretraining task akin to bidirectional modeling used to pretrain non-autoregressive BERT-style models (Devlin et al., 2019) for enhanced representation learning (Zhang et al., 2025b) and *label-free* transfer learning. Once the model weights listed up to Line 10 of Algorithm 1 have been (pre)trained, for finetuning, we freeze these weights and enable training of the $M$-layer auxiliary Transformer trunk weights following Line 10 using trunk layer $K$'s frozen embeddings. By default, $K = L$, which we ablate in Section 4.

**Predictive Losses.** For property prediction, we supervise the task's embeddings $p_b^{\text{props'}}$ using the mean absolute error (MAE) over each batch element $b$ as follows:

$$L_{\text{MAE}}(\boldsymbol{P}^{\text{props'}}) = \frac{1}{B} \sum_b \left| p_b^{\text{props'}} - p_b^{\text{props}} \right|, \tag{7}$$

where $B$ is the number of samples in a given mini-batch, $p_b^{\text{props'}}$ denotes the predicted property for the $b$-th sample, and $p_b^{\text{props}}$ denotes the true property for the same sample. Similarly, energy embeddings $p_b^{\text{energy'}}$ are supervised using the mean squared error (MSE) over each batch element:

$$L_{\text{MSE}}(\boldsymbol{P}^{\text{energy'}}) = \frac{1}{B} \sum_b \left( p_b^{\text{energy'}} - p_b^{\text{energy}} \right)^2. \tag{8}$$

Lastly, force embeddings $p_b^{\text{forces'}}$ are also learned using a batch-wise MSE loss, with the loss weight $\lambda_{\text{forces}}$:

$$L_{\text{MSE}}(\boldsymbol{P}^{\text{forces'}}) = \mathbb{E}\left[ \frac{1}{N} \| \boldsymbol{P}^{\text{forces'}} - \boldsymbol{P}^{\text{forces}} \|_2^2 \right]. \tag{9}$$

## 3 EXPERIMENTAL SETUP

**Datasets.** Following Joshi et al. (2025), we train models on periodic materials from MP20 and non-periodic molecules from QM9 for our generative pretraining experiments. MP20 (Xie et al., 2022) is a collection of 45,231 metastable crystal structures from the Materials Project (Jain et al., 2013), with up to 20 atoms in each material's unit cell spanning 89 different atom/element types. QM9 (Wu et al., 2018) is comprised of 130,000 stable small molecules with up to 9 heavy atoms (C, N, O, F) along with hydrogens. To ensure fair comparisons, we split the data according to prior work (Xie et al., 2022; Hoogeboom et al., 2022). Additionally, we report generative pretraining results for GEOM-Drugs (Axelrod & Gomez-Bombarelli, 2022), a collection of 430,000 large molecules containing up to 180 atoms, following the benchmarking setup of Vignac et al. (2023). Lastly, in Appendix E, we employ the QMOF dataset of 14,000 metal-organic framework (MOF) structures (Rosen et al., 2021) after removing structures containing more than 150 atoms, following Joshi et al. (2025).

For predictive finetuning, we use the QM9 and Matbench datasets, the latter supporting several fundamental property prediction tasks in materials science (Dunn et al., 2020). Further, in Appendix E, we adopt the OMol25 (4M) (Levine et al., 2025) and MPtrj (Deng et al., 2023) datasets for molecule and materials energy and force prediction, respectively. Data splits follow prior work for fair evaluation (Liao & Smidt, 2023; Dunn et al., 2020; Levine et al., 2025; Deng et al., 2023).

**Training and hyperparameters.** Our generative pretraining procedure for each model is described in Tables 7 and 8 of Appendix C. Predictive finetuning is outlined in Table 9 of Appendix C, during which we finetune the weights dedicated to a particular task individually to maximize training batch throughput and minimize time to convergence.

Key inference hyperparameters include the white noise scale $\gamma_g$ of each modality (following Geffner et al. (2025)) and the number of integration steps $T$. Through an extensive hyperparameter sweep illustrated in Figure 5 of Appendix C, we find that $T = 50$ or $100$ with $\gamma_g = 0.01$ generally works well for molecule and material generation, except for QM9-sized small molecules, which benefit from adding additional white noise (e.g., 50) to non-periodic atom positions to improve sample uniqueness/diversity. Like Vonessen et al. (2025), $g(t) = \frac{1}{t+0.01}$ for noise scaling by default.

**Evaluation metrics.** Following previous work (Hoogeboom et al., 2022; Joshi et al., 2025), for generative pretraining, we evaluate ZATOM-1's ability to generate valid and realistic molecules and materials by sampling 10,000 molecules and 2,500 materials and computing validity, uniqueness, and PoseBusters sanity checks for molecules (Buttenschoen et al., 2024) and calculating validity, uniqueness, novelty, energy, and stability metrics using LeMat-GenBench (Duval et al., 2025). These generative metrics are described in further detail in Appendix D. For predictive finetuning, we assess ZATOM-1's MAE for property, energy, and force prediction tasks, scaled/unit-converted according to literature benchmarking conventions.

**Baselines.** For generative pretraining, we compare ZATOM-1 trained jointly on MP20 and QM9 to material-only and molecule-only variants, as well as state-of-the-art baselines for both data sources trained in one or two-stage configurations. Generated samples are shown in Appendix F.

Table 1: **Crystal generation results on LeMat-GenBench with MP20 training.** Core model evaluation metrics are calculated for 2,500 sampled crystals using an *MLIP ensemble* with *LeMat-Bulk* as the reference set. Results are organized into three categories separated by horizontal lines: (1) for models that incorporate *relaxation* as part of their generation pipeline (MatterGen, PLaID++, etc.); (2) for other baseline generative models; and (3) for our proposed generative models. Best values are shown in **bold**, second-best values are underlined. Notably, jointly trained ZATOM-1 (80M) achieves its results using ∼400 GPU training hours, 3x faster than jointly trained ADiT (180M).

| Model | # Params | Valid% ↑ | Unique% ↑ | Novel% ↑ | Energy-based | | | Stability | | Metastability | |
|---|---|---|---|---|---|---|---|---|---|---|---|
| | | | | | $E_f$ (Std) ↓ | $E_{hull}$ (Std) ↓ | RMSD (Std) ↓ | Stable% ↑ | SUN% ↑ | Metastable% ↑ | MSUN% ↑ |
| **With relaxation** | | | | | | | | | | | |
| MatterGen | 47M | 95.7 | **95.1** | 70.5 | −0.70 ± 0.79 | 0.18 ± 0.18 | 0.39 ± 0.50 | 2.0 | 0.2 (6) | 33.4 | **15.0** |
| PLaID++ | 7B | **96.0** | 77.8 | 24.2 | −0.50 ± 0.44 | 0.09 ± 0.16 | **0.13 ± 0.29** | **12.4** | **1.0 (26)** | 60.7 | 7.6 |
| WyFormer | 8B | 93.4 | 93.0 | 66.4 | −0.43 ± 0.95 | 0.50 ± 0.51 | 0.81 ± 0.98 | 0.5 | 0.1 (3) | 15.7 | 1.9 |
| WyFormer-DFT | 8B | 95.2 | 95.0 | 66.4 | −0.67 ± 0.91 | 0.27 ± 0.36 | 0.42 ± 0.60 | 3.7 | 0.4 (9) | 24.8 | 7.8 |
| **Without relaxation** | | | | | | | | | | | |
| Jointly trained ADiT | 180M | 90.6 | 87.8 | 26.0 | −0.73 ± 0.93 | 0.33 ± 0.45 | 0.38 ± 0.40 | 0.4 | 0.0 (0) | 36.5 | 1.0 |
| Crystal-GFN | 1M | 51.7 | 51.7 | 51.7 | **−1.30 ± 2.63** | 2.09 ± 2.38 | 1.87 ± 0.97 | 0.0 | 0.0 (0) | 0.0 | 0.0 |
| Crystalformer | 5M | 69.9 | 69.4 | 31.8 | −0.17 ± 1.48 | 0.70 ± 1.33 | 0.66 ± 1.05 | 1.4 | 0.0 (0) | 28.8 | 3.1 |
| DiffCSP | 12M | 95.7 | 94.8 | 66.2 | −0.64 ± 0.96 | 0.28 ± 0.56 | 0.59 ± 0.63 | 2.3 | 0.1 (3) | 29.8 | 8.5 |
| DiffCSP++ | 12M | 95.3 | **95.1** | 62.0 | −0.52 ± 0.87 | 0.41 ± 0.44 | 0.69 ± 0.82 | 1.0 | 0.2 (4) | 26.4 | 5.0 |
| LLaMat2-CIF | 7B | 84.4 | 81.4 | 30.0 | −0.47 ± 1.01 | 0.44 ± 0.71 | 0.54 ± 0.71 | 0.7 | 0.1 (3) | 34.7 | 2.1 |
| LLaMat3-CIF | 8B | 15.4 | 15.2 | 10.5 | 0.74 ± 2.69 | 1.71 ± 2.48 | 1.01 ± 1.10 | 0.1 | 0.0 (0) | 2.1 | 0.2 |
| SymmCD | 12M | 73.4 | 73.0 | 47.0 | −0.02 ± 1.22 | 0.88 ± 1.07 | 0.87 ± 1.09 | 1.4 | 0.1 (2) | 18.6 | 2.4 |
| **Ours: Without relaxation** | | | | | | | | | | | |
| MP20-only ZATOM-1 | 80M | 73.2 | 70.4 | 21.0 | −0.53 ± 0.97 | 0.44 ± 0.64 | 0.42 ± 0.49 | 0.6 | 0.00 (0) | 32.7 | 0.20 (5) |
| Jointly trained ZATOM-1 | 80M | 88.5 | 84.4 | 8.1 | −0.84 ± 0.89 | 0.19 ± 0.37 | 0.21 ± 0.27 | 2.7 | 0.04 (1) | 51.8 | 0.36 (9) |
| Jointly trained ZATOM-1-L | 160M | 95.0 | 90.4 | 3.7 | −0.87 ± 0.86 | **0.09 ± 0.21** | 0.14 ± 0.20 | 3.1 | 0.04 (1) | **71.5** | 0.64 (16) |
| Jointly trained ZATOM-1-XL | 300M | 94.9 | 90.1 | 4.8 | −0.91 ± 0.87 | 0.11 ± 0.23 | 0.16 ± 0.19 | 3.3 | 0.00 (0) | 64.3 | 0.36 (9) |

Table 2: **Molecule generation results on QM9.** We report (a) validity and uniqueness rates as well as (b) % pass rates on 7 sanity checks from PoseBusters for 10,000 sampled molecules. All models explicitly generate hydrogen atoms. Notably, jointly trained ZATOM-1 (80M) achieves its results using ∼400 GPU training hours compared to jointly trained ADiT (180M), which requires ∼1,200 GPU pretraining + finetuning hours.

**(a) Validity results**

| Model | # Params | Validity (%) ↑ | Unique (%) ↑ |
|---|---|---|---|
| **One-stage training** | | | |
| Equivariant Diffusion | 20M | 91.90 | 98.69 |
| Symphony | 2M | 83.50 | 97.98 |
| **Two-stage training** | | | |
| GeoLDM | 20M | 93.80 | 98.82 |
| QM9-only ADiT | 180M | 92.19 | 97.90 |
| Jointly trained ADiT | 180M | 94.45 | 97.82 |
| **Ours: One-stage training** | | | |
| QM9-only ZATOM-1 | 80M | 92.88 | 97.71 |
| Jointly trained ZATOM-1 | 80M | 94.94 | 97.16 |
| Jointly trained ZATOM-1-L | 160M | 95.26 | 96.84 |
| Jointly trained ZATOM-1-XL | 300M | 95.19 | 97.10 |

**(b) PoseBusters results**

| Test (% pass) ↑ | Symphony | Eq. Diff. | ADiT | ZATOM-1 |
|---|---|---|---|---|
| Atoms connected | 99.92 | 99.88 | 99.70 | 99.98 |
| Bond angles | 99.56 | 99.98 | 99.85 | 99.95 |
| Bond lengths | 98.72 | 100.00 | 99.41 | 99.97 |
| Aromatic ring flat | 100.00 | 100.00 | 100.00 | 100.00 |
| Double bond flat | 99.07 | 98.58 | 99.98 | 99.99 |
| Internal energy | 95.65 | 94.88 | 95.86 | 99.78 |
| No steric clash | 98.16 | 99.79 | 99.79 | 99.81 |

Our experiments for materials generation using LeMat-GenBench include comparisons to: (1) six equivariant/invariant generative models adopting handcrafted multimodal product manifolds: MatterGen (Zeni et al., 2025), Crystal-GFN (Hernández-García et al., 2023), DiffCSP (Jiao et al., 2023), DiffCSP++ (Jiao et al., 2024), SymmCD (Levy et al., 2025), and FlowMM (Miller et al., 2024); (2) ADiT (Joshi et al., 2025), a non-equivariant molecule & material flow matching model that learns symmetries from data; and (3) four language model-based generative methods for materials: PLaID++ (Xu et al., 2025), WyFormer (Kazeev et al., 2025), Crystalformer (Cao et al., 2025), and LLaMat (Mishra et al., 2024).

Our molecule generation experiments compare ZATOM-1 to: (1) four generative methods operating on an equivariant multimodal product manifold: Equivariant Diffusion (Hoogeboom et al., 2022), Symphony (Daigavane et al., 2024), EQGAT-diff (Le et al., 2024), and SemlaFlow (Irwin et al., 2025); (2) two latent diffusion/flow matching generative models: GeoLDM (Xu et al., 2023) and ADiT (Joshi et al., 2025); and (3) non-equivariant TABASCO (Vonessen et al., 2025) which learns symmetries from data.

For predictive finetuning, we compare ZATOM-1 to: (1) 11 highly-optimized property prediction methods: DimeNet++ (Gasteiger et al., 2020), EGNN (Satorras et al., 2021), PaiNN (Schütt et al., 2021), TorchMD-NET (Thölke & Fabritiis, 2022), SphereNet (Liu et al., 2022b), SEGNN (Brandstetter et al., 2022), EQGAT (Le et al., 2022), Equiformer (Liao & Smidt, 2023), EquiformerV2 (Liao et al., 2024), PΘNITA (Bekkers et al., 2024), and the Platonic Transformer (Islam et al., 2025); (2) an unoptimized single-task property prediction method: AIM-STL (Minot & Schneider, 2025); and (3)

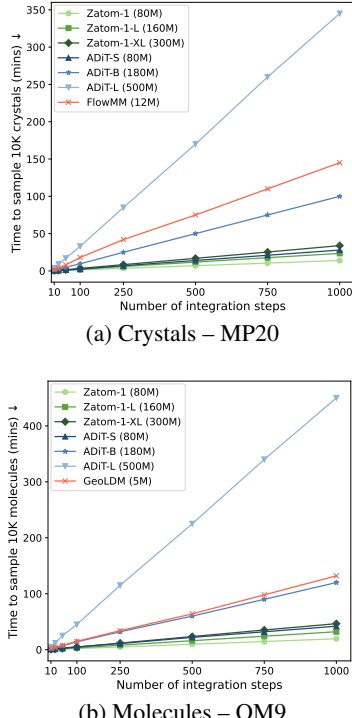

(a) Crystals – MP20

(b) Molecules – QM9

Figure 2: **ZATOM-1 is significantly faster than equivariant and latent diffusion models.** We plot the number of integration steps for ZATOM-1, equivariant diffusion FlowMM, and latent diffusion ADiT vs. time to generate 10,000 samples on a single NVIDIA GPU. ZATOM-1 scales significantly better with the number of integration steps compared to equivariant and latent diffusion models.

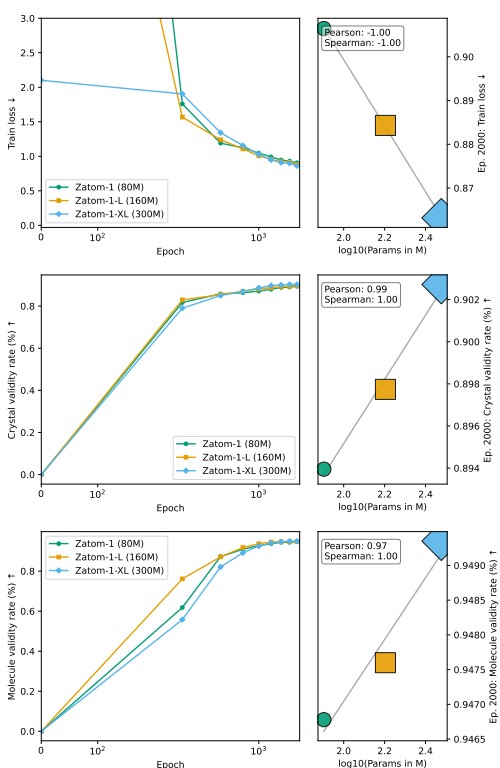

Figure 3: **Scaling up ZATOM-1 predictably improves performance.** We show the effect of increasing the number of ZATOM-1 model parameters on the training loss and generation validity rates. **Left:** training loss and validity rates vs. epochs. **Right:** correlation plots for training loss and validity rates at epoch 2,000 vs. ZATOM-1 parameters (in Millions).

two unoptimized multi-task property prediction methods: AIM-MTL-Scalar and AIM-MTL-Matrix (Minot & Schneider, 2025). Lastly, our energy and force prediction experiments for molecules and materials compare ZATOM-1 to the equivariant eSEN (Fu et al., 2025) and Orb-v1 (Neumann et al., 2024) models, respectively.

## 4 RESULTS

**State-of-the-art molecule (crystal) generation.** Tables 1 and 2 show that ZATOM-1 benefits from positive transfer learning between chemical domains compared to MP20-only or QM9-only training. In particular, Table 1's LeMat-GenBench results demonstrate that ZATOM-1 can generate highly stable, unique, and novel (SUN) materials after rigorous sample evaluation, whereas previous flow matching-based methods, such as ADiT, cannot. Nonetheless, as Duval et al. (2025) have suggested, ZATOM-1's novelty and SUN rates and model scaling benefits could likely be improved by training on a larger set of diverse 3D materials, such as (5M) LeMat-Bulk (Siron et al., 2025), in future work. Table 2's QM9 results highlight that ∼99% of ZATOM-1's generated molecules pass all of the PoseBusters sanity checks, while ∼95% of ADiT's molecules satisfy each check, establishing a new state-of-the-art. In line with Table 2's results, Table 3 shows that ZATOM-1 is the first generative model to reach a (maximum-possible) PoseBusters validity rate of 94% for larger molecules in GEOM-Drugs (Vonessen et al., 2025), demonstrating the strength of ZATOM-1's molecule & material pretraining for generating large, drug-like molecules. Important to note is that, due to QM9 (MP20)'s

Table 3: **Molecule generation results on GEOM-DRUGS.** *Left:* Validity, uniqueness, and % pass rates on PoseBusters for 10,000 sampled molecules. ZATOM-1 matches or exceeds state-of-the-art equivariant/latent diffusion baselines and generates more PoseBusters-valid samples than the recent baseline TABASCO. *Right:* We plot the number of integration steps for ZATOM-1, (latent diffusion) ADiT, and (equivariant) SemlaFlow vs. time to generate 10,000 molecules on a single NVIDIA GPU. ZATOM-1 scales competitively with the number of integration steps compared to SemlaFlow/ADiT.

| Metric (% pass) ↑ | EQGAT-diff | SemlaFlow | ADiT | TABASCO | ZATOM-1 | GEOM-Drugs |
|---|---|---|---|---|---|---|
| Validity | 94.6 | 93.9 | 95.3 | 97.6 | 93.6 | - |
| Uniqueness | 100.0 | 100.0 | 100.0 | 99.03 | 99.93 | - |
| Atoms connected | 84.4 | 92.3 | 93.0 | 99.9 | 99.6 | - |
| Bond angles | 86.9 | 94.8 | 92.3 | 99.2 | 99.6 | - |
| Bond lengths | 87.0 | 94.6 | 92.5 | 99.4 | 99.7 | - |
| Aromatic ring flat | 87.0 | 94.9 | 95.4 | 100.0 | 100.0 | - |
| Double bond flat | 87.0 | 94.2 | 95.3 | 99.8 | 100.0 | - |
| Internal energy | 86.8 | 94.8 | 91.3 | 99.4 | 99.3 | - |
| No steric clash | 82.9 | 92.0 | 91.8 | 94.3 | 96.4 | - |
| PoseBusters valid | 59.7 | 87.5 | 85.3 | 91.6 | 94.1 | 94.0 |

small size, for molecule (material) generation, increasing model size exchanges the model's rate of sample uniqueness (novelty) for increased sample validity (stability). For interested readers, we report results for O(3)-equivariant PLATOM-1 in Appendix B.

**ZATOM-1 is faster than baselines.** For both QM9 and MP20 sample generation, Figure 2 illustrates that ZATOM-1 offers an order of magnitude faster inference speeds than ADiT, a state-of-the-art baseline, and Table 3 shows that ZATOM-1's speeds scale better for GEOM-Drugs sampling.

**ZATOM-1 improves with scale.** Figure 3 highlights that ZATOM-1's training and validation metrics improve predictably as the model's size increases, suggesting that further performance improvements may be obtained with parallel data and model scaling (Kaplan et al., 2020).

Table 4: **Property prediction for 3D molecules (QM9) / materials (Matbench, zero-shot).** Results: *test* mean absolute errors (std. dev. over three runs with random seeds); †: using different data partitions; /: a joint molecule & material property prediction task. Best (second-best) results for each (optimized or unoptimized) category are in **bold** (underlined). Notably, ZATOM-1 (80M ❄ + 20M 💧) and AIM-MTL are the only multi-task learning (MTL) methods represented in these results, which they achieved with only ∼100 GPU hours of additional finetuning for molecule property prediction. In contrast, single-task learning (STL) baselines such as EquiformerV2 (11M) traditionally require ∼150 GPU training hours *for each task*, due to their extensive hyperparameter optimization per task.

| Model / Metrics ↓ | # Params | $\alpha$ ($a_0^3$) | $\Delta\varepsilon$ (meV) | $\varepsilon_{HOMO}$ (meV) | $\varepsilon_{LUMO}$ (meV) | $\mu$ (D) | $C_v$ (cal/mol K) | $G^{ATOM}$ (meV) | $H^{ATOM}$ (meV) | $R^2$ ($a_0^2$) | $U^{ATOM}$ (meV) | $U_0^{ATOM}$ (meV) | $ZPVE$ (meV) |
|---|---|---|---|---|---|---|---|---|---|---|---|---|---|
| **Optimized single-task learning** | | | | | | | | | | | | | |
| DimeNet++† | 5M | .044 | 33.0 / **199** | 25 | 20 | .030 | .023 | 8.00 | 7.00 | .331 | 6.00 | 6.00 | 1.21 |
| EGNN† | 1M | .071 | 48.0 / — | 29 | 25 | .029 | .031 | 12.0 | 12.0 | .106 | 12.0 | 11.0 | 1.55 |
| PaiNN | 1M | .045 | 46.0 / — | 28 | 20 | .012 | .024 | 7.35 | 5.98 | .066 | 5.83 | 5.85 | 1.28 |
| TorchMD-NET | 7M | .059 | 36.0 / — | 20 | 18 | .011 | .026 | 7.62 | 6.16 | **.033** | 6.38 | 6.15 | 1.84 |
| SphereNet | 2M | .046 | 32.0 / — | 23 | 18 | .026 | **.021** | 8.00 | 6.00 | .292 | 7.00 | 6.00 | **1.12** |
| SEGNN† | 1M | .060 | 42.0 / — | 24 | 21 | .023 | .031 | 15.0 | 16.0 | .660 | 13.0 | 15.0 | 1.62 |
| EQGAT | - | .053 | 32.0 / — | 20 | 16 | .011 | .024 | 23.0 | 24.0 | .382 | 25.0 | 25.0 | 2.00 |
| Equiformer | 4M | .046 | 30.0 / — | 15 | 14 | .011 | .023 | 7.63 | 6.63 | .251 | 6.74 | 6.59 | 1.26 |
| EquiformerV2 | 11M | .050 | **29.0** / — | **14** | **13** | .010 | .023 | 7.57 | 6.22 | .186 | 6.49 | 6.17 | 1.47 |
| P6NITA | - | **.038** | 30.4 / — | 16 | 15 | .012 | .024 | 8.63 | 8.04 | .235 | 8.67 | 8.31 | 1.29 |
| Platonic Transformer | - | .049 | 37.4 / — | 22 | 17 | **.010** | .024 | 12.0 | 12.0 | .222 | 11.9 | 13.0 | 1.30 |
| **Unoptimized single-task learning** | | | | | | | | | | | | | |
| AIM-STL† | - | .181 | — / — | 61 | 54 | **.067** | .072 | 66.2 | 63.9 | **.503** | 64.2 | 58.8 | 4.54 |
| **Unoptimized multi-task learning** | | | | | | | | | | | | | |
| AIM-MTL-Scalar† | - | .268 | — / — | 59 | 71 | .089 | .103 | 113 | 112 | 4.18 | 112 | 112 | 9.82 |
| AIM-MTL-Matrix† | - | .251 | — / — | 61 | 72 | .088 | .103 | 112 | 118 | 4.07 | 116 | 116 | 12.2 |
| Non-pretrained ZATOM-1 (OURS) | 80M ❄ + 20M 💧 | .140±.001 | 72.3±0.1 / 3358±6 | 54±0.0 | 49±0.0 | .157±.001 | .064±.000 | 45.5±0.2 | 44.7±0.2 | 4.72±0.02 | 45.2±0.2 | 45.8±0.2 | 2.89±0.01 |
| QM9-pretrained ZATOM-1 (OURS) | 80M ❄ + 20M 💧 | .095±.001 | 49.5±0.3 / 5297±29 | 36±0.2 | 34±0.1 | .100±.000 | .044±.000 | **23.1**±0.2 | 23.1±0.2 | 3.63±0.01 | **23.5**±0.2 | **22.6**±0.2 | 2.04±0.01 |
| Jointly pretrained ZATOM-1, $K = L/2$ (OURS) | 80M ❄ + 20M 💧 | .099±.001 | 51.2±0.3 / **2640±8** | 37±0.1 | 34±0.2 | .121±.001 | .044±.000 | 24.1±0.1 | 23.3±0.2 | 3.96±0.01 | 23.5±0.2 | 23.4±0.1 | 2.01±0.01 |
| Jointly pretrained ZATOM-1-XL (OURS) | 300M ❄ + 20M 💧 | .104±.000 | 53.1±0.1 / 3478±14 | 38±0.1 | 37±0.1 | .110±.000 | .046±.000 | 26.6±0.0 | 26.2±0.1 | 3.75±0.02 | 29.2±0.0 | 26.6±0.1 | 2.14±0.01 |
| Jointly pretrained ZATOM-1 (OURS) | 80M ❄ + 20M 💧 | **.091**±.001 | **46.2**±0.2 / 3891±20 | **34**±0.1 | **32**±0.2 | .090±.001 | **.041**±.000 | 23.3±0.4 | **23.0**±0.4 | 3.29±0.01 | 24.0±0.4 | 23.3±0.4 | **1.98**±0.01 |

**Chemical transfer learning improves predictions.** Table 4 demonstrates that ZATOM-1 achieves state-of-the-art *multi-task* property prediction performance for QM9. For the majority of QM9's target properties, these results further mark the *first* instance of positive predictive transfer learning and zero-shot generalization via joint generative pretraining in SciML, indicating that the final trunk layer of ZATOM-1 excels in 3D molecular tasks ($K = L$), while its middle trunk layers are better suited for (zero-shot) 3D material tasks ($K = L/2$). Interestingly, increasing the size of ZATOM-1's pretrained trunk does not guarantee improvements for property prediction, which may be due to the modest size of ZATOM-1's pretraining datasets.

**Additional results.** We refer interested readers to Appendix E for the results of additional experiments, including those for metal-organic framework generation, materials finetuning for property prediction, and energy and force prediction towards machine learning-based interatomic potentials (MLIPs) (Wood et al., 2025). Notably, we demonstrate promising performance for materials force prediction compared to Orb-v1 and further find that ZATOM-1 can effectively balance (supervised) molecular and material property prediction with additional cross-domain finetuning.

## 5 CONCLUSION

In this work, we introduced ZATOM-1, the first general-purpose foundation model for 3D molecules and materials. Empirical results show that ZATOM-1 achieves state-of-the-art molecule generation and competitive materials generation, with fast training and inference speeds. Additionally, it demonstrates the advantages of generative pretraining for downstream prediction tasks, marking SciML's first instance of positive predictive transfer learning through joint generative pretraining. As a single neural network capable of multiple fundamental tasks in molecular and material science, to advance the development of atomic foundation models, ZATOM-1's code and weights are freely available at `https://github.com/Zatom-AI/zatom`.

## ACKNOWLEDGMENTS

This research used resources of the National Energy Research Scientific Computing Center, a DOE Office of Science User Facility supported by the Office of Science of the U.S. Department of Energy under Contract No. DE-AC02-05CH11231 using AI4Sci@NERSC award NERSC DDR-ERCAP0036206 awarded to AM. NBE would like to acknowledge support from the U.S. Department of Energy, Office of Science, Office of Advanced Scientific Computing Research, EXPRESS: 2025 Exploratory Research for Extreme-Scale Science program, and the Scientific Discovery through Advanced Computing (SciDAC) program, under Contract Number DE-AC02-05CH11231 at Berkeley Lab.

## IMPACT STATEMENT

This paper presents work intended to advance the field of Scientific Machine Learning. There are many potential societal consequences of our work, including the acceleration of scientific discovery in the molecular and materials sciences. We acknowledge the risk that, in the hands of "bad actors", such technologies may be used with harmful ends in mind. However, it is our hope that efforts in developing scientific foundation models for the chemical sciences will disproportionately influence the positive societal outcomes of such research, such as improved medicines and energy technologies, as opposed to possible negative consequences, such as the development of new bioweapons.

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

APPENDICES

# A   RELATED WORK

The concept of scientific foundation models, which are large, self-supervised networks pretrained on broad scientific data, has recently gained momentum in chemistry and materials science (Pyzer-Knapp et al., 2025; Choi et al., 2025; Edamadaka et al., 2025). Recent reviews emphasize the value of training on diverse chemical datasets and using generative or multi-task objectives to build broadly useful chemical representations (Pyzer-Knapp et al., 2025; Choi et al., 2025; Wadell et al., 2025). However, most existing works remain domain-specific: "foundation" efforts in molecules often focus on SMILES or 2D graph pretraining, while material science approaches largely target property prediction or synthesis planning rather than joint 3D modeling.

**Generative modeling of 3D molecules.**   Several recent works propose equivariant diffusion or flow models for de novo 3D molecule design. Hoogeboom et al. (2022) introduce an E(3)-equivariant diffusion model (EDM) that denoises continuous atom coordinates and types together. Autoregressive graph models like Symphony (Daigavane et al., 2024) build molecules fragment-by-fragment with spherical harmonics features. Le et al. (2024) explore E(3)-equivariant diffusion losses (EQGAT-diff) to improve validity on QM9 and GEOM-Drugs. Latent diffusion approaches like GeoLDM (Xu et al., 2023) encode 3D structures into continuous latent spaces. More recent work explores the importance of equivariance: for instance, SemlaFlow uses flow matching with an equivariant graph neural network backbone (Irwin et al., 2025); FlowMol3 employs hybrid continuous-discrete flow matching with chirality-aware message passing (Dunn & Koes, 2025); and TABASCO shows that a non-equivariant transformer can match the validity of equivariant models while generating 3D drug-like molecules (Vonessen et al., 2025). In summary, these models each focus on molecule generation but do not generalize across materials.

**Generative modeling of 3D materials.**   Likewise, generative models for periodic crystals have been developed separately. Score-based models (diffusion) and flow matching have been applied to crystal generation: e.g., Xie et al. (2022) propose CDVAE, a score-based model for generating periodic unit cells, and Jiao et al. (2023) introduce DiffCSP, an E(3)-equivariant diffusion model for predicting stable crystal structures. Recent work often uses variants of normalizing flows: Miller et al. (2024)'s FlowMM applies Riemannian flow matching over atom types and lattice geometry, and Yang et al. (2024) show that standard (non-equivariant) diffusion on a learnable crystal representation (UniMat) can scale to large collections of structures. Another parallel thread is generative models for inorganic materials like those of Zeni et al. (2025). FlowLLM combines a fine-tuned language model base with flow matching for crystals (Sriram et al., 2024), while the All-Atom Diffusion Transformer (ADiT) unifies molecule and material generation by learning a shared latent autoencoding of both domains and diffusing in that latent space (Joshi et al., 2025). Notably, none of these approaches simultaneously addresses both molecules and materials in one model without resorting to complex two-stage training approaches, such as latent diffusion or LLM-based data generation, nor do they leverage generative pretraining for downstream tasks.

**Multimodal flow/diffusion architectures.**   Our work joins a growing class of multimodal generative models that handle discrete atom types and continuous 3D geometries jointly. For example, Miller et al. (2024) explicitly matches flows over categorical atom labels, continuous coordinates, and lattice parameters in a single framework. ADiT (Joshi et al., 2025) analogously embeds all-atom molecular and crystal representations into a shared latent space for latent diffusion. These approaches stand in contrast to earlier graph or point-based generative models (e.g., EDM (Hoogeboom et al., 2022), EQGAT-diff (Le et al., 2024)) that typically model only a small number of data modalities simultaneously. By jointly training on atom types, positions, and unit cell parameters, our approach builds on the multimodal flow concept of FlowMM while using a standard Transformer backbone for expressiveness, scalability, and speed.

**Representation learning and pretraining in chemistry.**   Beyond full generative models, chemical representation learning has employed self-supervised pretraining on molecular graphs. For instance, GraphMVP (Liu et al., 2022a) uses contrastive learning between 2D molecular topology and 3D geometry. Masked autoencoder approaches and property prediction pretraining have also been explored. However, most existing pretraining does not integrate full sample generation. Even though image diffusion models have been shown to yield useful feature embeddings (Li et al., 2023; Vélez

et al., 2025), few works have investigated generative pretraining as a widely applicable paradigm in 3D chemistry or AI for Science more broadly. Accordingly, our work frames conditional multimodal flow matching (over atom types and geometry) as a self-supervised "pretraining" task for 3D chemical systems. By pretraining on joint molecule & material data, we obtain unified embeddings transferable to property, energy, and force prediction. This idea is similar to Chang & Ye (2024)'s structure-property molecular model, which learns a bidirectional generative model of molecular graphs and properties, but here we apply it broadly across both molecules and crystals.

**Overview.** Overall, prior work in molecular/material generation has focused either on discrete (graph/SMILES) models or on separate generative flows/diffusions for molecules or crystals (Hoogeboom et al., 2022; Xie et al., 2022). Our approach differs by unifying generative and predictive modeling for all-atom 3D chemistry. We draw on advances in flow matching and multimodal architecture design (Song et al., 2023; Miller et al., 2024; Vonessen et al., 2025), and we treat generative modeling itself as a pretraining strategy. This differs from previous scientific foundation models that often focus on language or graph representations (Pyzer-Knapp et al., 2025; Choi et al., 2025; Wadell et al., 2025), or that rely on autoregression and do not provide end-to-end experimental reproducibility (Zhang et al., 2025a). By explicitly modeling the ambient 3D space of atoms and elements together, we extend the flow-based generative paradigm to create a single foundation model across 3D molecular and material domains.

# B  PLATOM-1 – EXPLICITLY EQUIVARIANT TRANSFORMER ARCHITECTURE

A long-standing discussion in the molecular modeling community revolves around whether symmetries in the data should be learned or hard-coded into the model architecture. Given the considerable success of entirely non-equivariant models like AlphaFold3 (Abramson et al., 2024), the community's consensus has shifted toward the conclusion that equivariant models would be slower, harder to optimize, and generally inferior compared to architectures without such inductive biases. This appendix provides empirical evidence challenging this common belief, showing that – *when done right* – equivariance can unlock significant performance gains.

Section B.1 introduces the Trunk-based Flow Platoformer (TFP), a rotation equivariant counterpart of the Trunk-based Flow Transformer (TFT) architecture underlying ZATOM-1. Empirical results, showing faster convergence and higher validation scores of TFP, are presented in Section B.2 and Figure 4. For now, we focus exclusively on molecular tasks, since the current fractional coordinate-based invariant embedding of materials used in ZATOM-1 is at odds with the equivariant inputs expected by TFP.

## B.1  TRUNK-BASED FLOW PLATOFORMER ARCHITECTURE (TFP)

The Trunk-based Flow Platoformer (TFP) is a rotation equivariant counterpart of our TFT architecture from Section 2.2, replacing its standard transformer layers with *Platonic Transformer* layers introduced by Islam et al. (2025). These layers are equivariant to discrete subgroups of roto-reflections $G < O(3)$ which are symmetries of Platonic solids like the tetrahedron ($|G| = 12$) or cube ($|G| = 24$) (Cesa et al., 2022).

Rather than relying on $O(3)$-irrep features (i.e., $O(3)$-irreducible representations), as most equivariant point cloud message passing neural networks do, the features of TFP are associated with group elements, essentially encoding molecular geometry as viewed from $|G|$ different frames of reference. Mathematically, this is known as a *regular $G$-representation* (Weiler et al., 2025). In code, a point cloud of $N$ features with $C$ channels is represented by a tensor of shape $(N, |G|, C)$ rather than the usual shape $(N, C)$. Such regular representation-valued features are more commonly used in vision models, most notably group convolutional networks (Cohen & Welling, 2016; Weiler et al., 2018; Bekkers et al., 2018).

**TFP architecture.** Before applying Platonic layers, inputs need to be lifted to regular $G$-representations. Scalars (e.g., atom types) are lifted to features that are invariant over the group axis. Vector-valued inputs, like Euclidean coordinates, lift to non-trivial features encoding directional information. Platonic linear layers satisfy a symmetry constraint which reduces their parameter

count via $|G|$-fold weight sharing. The original linear embedding layers of TFT are replaced by such Platonic linear layers, applied after $G$-lifts.

Platonic attention layers utilize common scaled dot-product attention (Vaswani et al., 2017), and are thus compatible with IO-efficient Flash Attention kernels (Dao, 2024). The group axis of regular $G$-features is thereby reshaped into the attention heads axis rather than channels, i.e., $(H \cdot |G|, N, C)$ instead of $(H, N, |G| \cdot C)$ – this has the advantage that attention scores remain regular representation-valued rather than becoming $G$-invariant due to contracting over $G$ (Islam et al., 2025). We use full softmax attention instead of the linear attention variants explored in the original Platonic Transformer paper, as these were required mainly to address training instabilities when encoding rotary positions in molecular modeling tasks.

A core component of the Platonic Transformer is a $G$-equivariant generalization of Rotary Position Embeddings (RoPE) (Su et al., 2024), which guarantees both translational and $G$-equivariance, i.e., approximate E(3)-equivariance. As we aim to stick as close as possible to the original TFT design without *relative* RoPE attention, we omit this feature, relying merely on the *absolute* embedding of Euclidean coordinates via lifting and Platonic linear layers discussed above. Note that this design choice breaks translational symmetries but preserves $G$-equivariance. Since we zero-center non-periodic 3D atom coordinates, this is not an issue in practice.

We reimplement and extend many other operations from the original Platonic Transformer code, including cross-attention layers, new normalization modules, embedding layers, and optimized Platonic linear layers. Given these modules, we implement TFP in analogy to the TFT design summarized in Algorithm 1. Final predictions are projected back from $G$-valued features to scalar and vector outputs. Our implementation is available as part of the ZATOM-1 code repository.

**Compute and parameter cost.** As TFP operates on tensors of shape $(N, |G|, C)$ with an additional group axis, one may wonder how its computational, memory, and parameter complexities compare to those of the TFT baseline. Given that regular features have $|G|C$ effective dimensions, it is common practice to reduce their channel count $C$ relative to the $D$ channels of TFT's shape $(N, D)$ features. The implied complexities for TFT's and TFP's linear and attention layers are summarized in Table 5. From these complexities, the following two heuristics to choose $C$ to build "matching" architectures arise (Weiler & Cesa, 2019):

*Compute-matching*: To match the compute and memory cost of TFP and TFT, one sets $C = D/|G|$, essentially sticking with the same effective number of channels but splitting them into group and channel axes. However, this implies $G$ times fewer trainable parameters in TFP.

*Parameter-matching*: Alternatively, one may match the number of parameters of TFP and TFT by setting $C = D/\sqrt{|G|}$. This leads to a $|G|$ fold increase in memory and Floating-point Operations Per Second (FLOPS).

As a middle ground between these two extremes, we choose $C = D/|G|^{2/3}$, resulting in $\sqrt{|G|}$ fewer parameters and $\sqrt{|G|}$ increased computational cost. For the tetrahedral group, $|G| = 12$ and $\sqrt{|G|} \approx 3.5$, such that a TFP analog to ZATOM-1 with 80M parameters has 23M parameters. Note that this model happens to be approximately compute-matched with ZATOM-1-XL, which has 300M parameters, i.e., 3.75 times the number of ZATOM-1's parameters.

## B.2 EMPIRICAL RESULTS

Based on TFP, we implement PLATOM-1, the Platonic group equivariant counterpart of ZATOM-1. Due to limited computational resources, we specifically evaluate the (orientation-preserving) tetrahedral group with $|G| = 12$ rotations (no reflections) and 23M parameters as discussed above. Since PLATOM-1 is only equivariant w.r.t. discrete rotations, we use the same continuous SO(3) rotational data augmentation as used for ZATOM-1.

**QM9 molecule generation.** We first compare to the QM9-only pretrained ZATOM-1 baseline flow model in Figure 4. It is evident from these plots that PLATOM-1 converges much faster than ZATOM-1 in terms of training and validation losses, as well as validation metrics. Such improved convergence is common for equivariant models, as they do not need to implicitly learn symmetries from data (Weiler

|  | Linear | $G$-linear | Attention | $G$-attention |
|---|---|---|---|---|
| Compute FLOPS | $\mathcal{O}(ND^2)$ | $\mathcal{O}(N|G|^2C^2)$ | $\mathcal{O}(N^2D)$ | $\mathcal{O}(N^2|G|C)$ |
| Memory | $\mathcal{O}(ND)$ | $\mathcal{O}(N|G|C)$ | $\mathcal{O}(ND)$ | $\mathcal{O}(N|G|C)$ |
| Parameters | $D^2$ | $|G|\cdot C^2$ | — | — |

Table 5: **Compute, memory, and parameter complexity of TFT and TFP.** TFT and TFP operate on tensors of shapes $(N, |G|, C)$ and $(N, D)$, respectively. Memory costs follow immediately from these shapes. Linear layers in TFT come with the usual quadratic number of FLOPS and parameters in their $D$ channels. TFP is internally leveraging `nn.Linear` layers as well, however, after flattening features to shape $(N, |G|\cdot C)$, implying a computational cost quadratic in $|G|$ as well. Equivariance imposes a symmetry constraint on the matrices (weight sharing), resulting in a $|G|$ times-enhanced parameter efficiency. TFT's attention layers have the usual computational complexity linear in channels $D$, while those of TFP are linear in $|G|C$. Neither introduces additional parameters.

et al., 2025; Zhdanov et al., 2024). The best TFP checkpoint on the validation set is reached after 249 epochs, instead of 399 epochs for TFT, partially offsetting TFP's increased computational cost.

Table 6 confirms that these findings translate to all test metrics calculated for these checkpoints. In particular, PLATOM-1 achieves better results than ZATOM-1, prior equivariant QM9-only pretrained models Equivariant Diffusion (Hoogeboom et al., 2022) and Symphony (Daigavane et al., 2024), and the much larger and slower latent diffusion model ADiT (Joshi et al., 2025). It even outperforms our 300M parameter model ZATOM-1-XL, despite the latter being jointly trained on molecules and materials.

**Joint molecule and material pretraining.** One of the main hypotheses of ZATOM-1 is the benefit of positive transfer learning when pretraining jointly on molecular and material modalities. A similar mutual benefit should, in principle, be observable for PLATOM-1. However, our current encoding of materials is inherently incompatible with Platom-1's equivariant design, leading to training instabilities. Specifically, materials are not represented in terms of their (vector-valued) Euclidean coordinates, but rather in terms of fractional coordinates and lattice basis lengths and angles. All these quantities are scalars, and therefore lift to features that are invariant over the $G$-axis. Being blind to the original Euclidean geometry expected by PLATOM-1, the model fails to converge.

While ZATOM-1 can learn to leverage fractional coordinates inputs, we believe that this is not the ideal (equivariant) data representation for PLATOM-1, since it obfuscates the Euclidean geometry of the materials. As part of future work, we plan to experiment with TFP using a unified Euclidean coordinate representation of molecules and materials, for which we expect to see positive transfer learning results in a PLATOM-1-style model.

Table 6: **Molecule generation results on QM9 with explicit equivariance.** We report (a) validity and uniqueness rates as well as (b) % pass rates on 7 sanity checks from PoseBusters for 10,000 sampled molecules. All models explicitly generate hydrogen atoms. Notably, QM9-only PLATOM-1 achieves its results using 249 training epochs compared to QM9-only ZATOM-1 (80M), which requires 399 training epochs to converge. Despite its smaller size and not being jointly pretrained on molecules and materials, PLATOM-1 is competitive with ADiT and ZATOM-1-XL.

(a) **Validity results**

| Model | # Params | Validity (%) ↑ | Unique (%) ↑ |
|---|---|---|---|
| **One-stage training** | | | |
| Equivariant Diffusion | 20M | 91.90 | 98.69 |
| Symphony | 2M | 83.50 | 97.98 |
| **Two-stage training** | | | |
| GeoLDM | 20M | 93.80 | 98.82 |
| QM9-only ADiT | 180M | 92.19 | 97.90 |
| Jointly trained ADiT | 180M | 94.45 | 97.82 |
| **Ours: One-stage training** | | | |
| QM9-only ZATOM-1 | 80M | 92.88 | 97.71 |
| QM9-only PLATOM-1 | 23M | 95.20 | 98.13 |
| Jointly trained ZATOM-1 | 80M | 94.94 | 97.16 |
| Jointly trained ZATOM-1-L | 160M | 95.26 | 96.84 |
| Jointly trained ZATOM-1-XL | 300M | 95.19 | 97.10 |

(b) **PoseBusters results**

| Test (% pass) ↑ | Symphony (QM9-only) | Eq. Diff. (QM9-only) | ADiT (jointly trained) | PLATOM-1 (QM9-only) | ZATOM-1 (QM9-only) | ZATOM-1 (jointly trained) |
|---|---|---|---|---|---|---|
| Atoms connected | 99.92 | 99.88 | 99.70 | 99.97 | 99.95 | 99.98 |
| Bond angles | 99.56 | 99.98 | 99.85 | 99.94 | 99.90 | 99.95 |
| Bond lengths | 98.72 | 100.00 | 99.41 | 99.96 | 99.89 | 99.97 |
| Aromatic ring flat | 100.00 | 100.00 | 100.00 | 100.00 | 100.00 | 100.00 |
| Double bond flat | 99.07 | 98.58 | 99.98 | 100.00 | 100.00 | 99.99 |
| Internal energy | 95.65 | 94.88 | 95.86 | 99.73 | 99.63 | 99.78 |
| No steric clash | 98.16 | 99.79 | 99.79 | 99.84 | 99.81 | 99.81 |

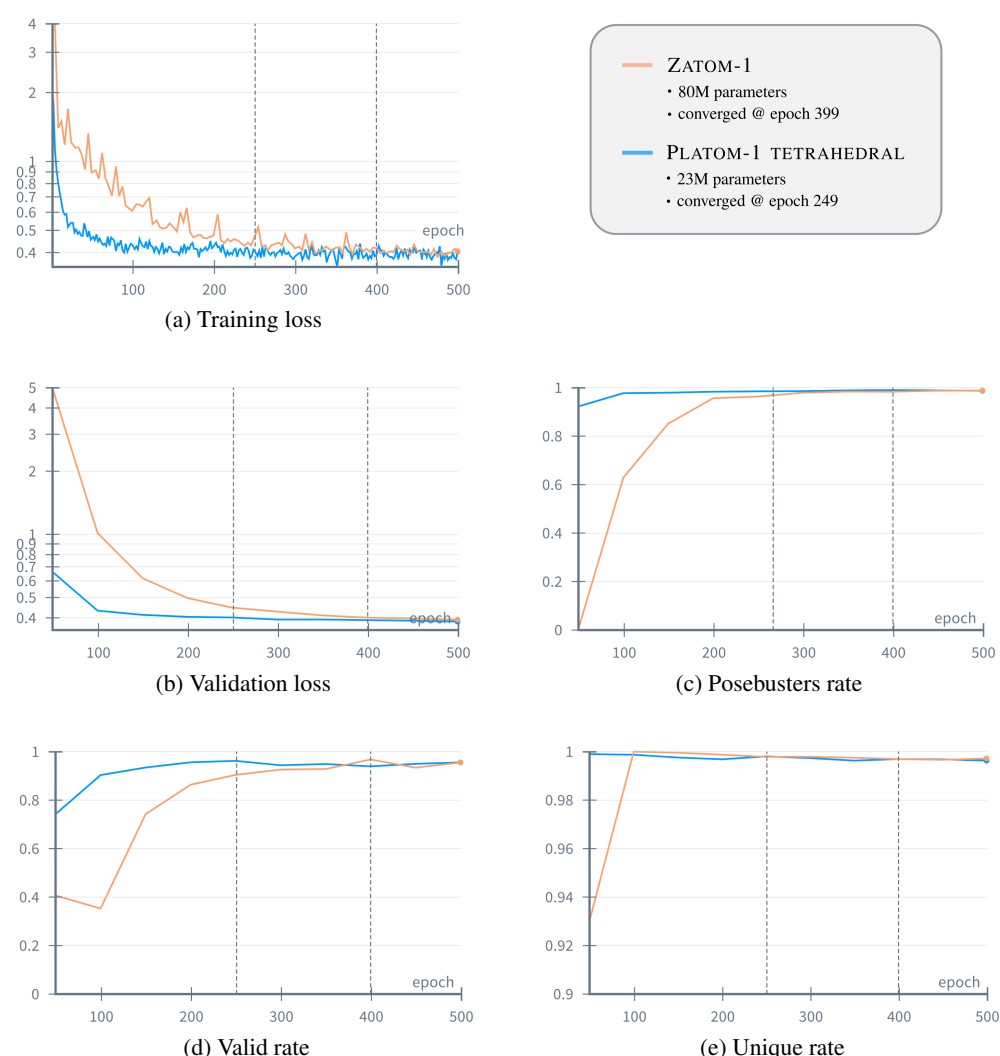

Figure 4: **QM9-only pretraining of Z**ATOM**-1 vs. P**LATOM**-1.** These plots show the training dynamics of ZATOM-1 compared to the analogous tetrahedral group equivariant PLATOM-1 model described in Section B.1. Due to its equivariance, the Platonic model does not explicitly need to learn symmetries, resulting in significantly faster convergence while achieving improved evaluation metrics. These findings translate to the test results in Table 6.

## C ADDITIONAL MODEL DETAILS

**Hyperparameters.** Table 7 reports the hyperparameters used to pretrain different sizes of ZATOM-1 on MP20 and QM9, and Table 8 shows which hyperparameters were used to pretrain ZATOM-1 on QMOF as well as on GEOM-Drugs jointly with other datasets. Likewise, Table 9 displays the hyperparameters used to finetune ZATOM-1 (80M) for the downstream tasks explored in this work.

For generative inference, Figure 5 shows the results of a hyperparameter sweep, where we find that using a limited number of integration steps and adding (varying levels of) white noise to each continuous modality during sampling generally improves sample validity and uniqueness. Notably, we find that for larger chemical systems, such as the molecules present in GEOM-Drugs, a smaller noise scale for atom positions (e.g., 0.01) is beneficial. Additionally, in all generative contexts, we unexpectedly find that classifier-free guidance worsens the quality of generated samples considerably (e.g., degrading 90% materials validity rates to 40%), motivating us to disable such inference guidance by default.

Table 7: Pretraining hyperparameters across different model sizes when training jointly on QM9 & MP20.

| Hyperparam. | ZATOM-1 (80M) | ZATOM-1-L (160M) | ZATOM-1-XL (300M) |
|---|---|---|---|
| # Params. | 77,350,264 | 162,615,544 | 293,887,096 |
| Hidden size | 512 | 768 | 1024 |
| # Transformer blocks | 16 | 16 | 16 |
| # Attn. heads | 8 | 8 | 8 |
| Train $t$ distribution | Beta(1.8,1) | Beta(1.8,1) | Beta(1.8,1) |
| $\lambda_{\text{discrete}}$ | 0.1 | 0.1 | 0.1 |
| Learning rate | 0.001 | 0.0005 | 0.00025 |
| Learning rate schedule | N/A | N/A | N/A |
| Optimizer | AdamW | AdamW | AdamW |
| Weight decay | 0.0 | 0.0 | 0.0 |
| EMA weight | 0.999 | 0.999 | 0.999 |
| Batch size | 256 | 128 | 64 |
| # Grad. accum. steps | 1 | 2 | 4 |
| # Rotation augs. | 8 | 8 | 8 |
| Effective batch size | 32,768 | 32,768 | 32,768 |
| # GPUs | 16 (80GB) | 16 (80GB) | 16 (80GB) |
| Training duration | 24h | 48h | 96h |
| # Sampling steps | 100 | 100 | 100 |
| $g(t)$ | $\frac{1}{t+0.01}$ | $\frac{1}{t+0.01}$ | $\frac{1}{t+0.01}$ |
| $\gamma_g$ ($\rightarrow$ for small molecule coordinates) | 0.01 ($\rightarrow$ 50.0) | 0.01 ($\rightarrow$ 50.0) | 0.01 ($\rightarrow$ 50.0) |

Table 8: Pretraining hyperparameters when training ZATOM-1 (80M) on GEOM-Drugs or QMOF.

| Hyperparam. | GEOM-Drugs & MP20 | QMOF |
|---|---|---|
| # Params. | 77,350,264 | 77,350,264 |
| Hidden size | 512 | 512 |
| # Transformer blocks | 16 | 16 |
| # Attn. heads | 8 | 8 |
| Train $t$ distribution | Beta(1.8,1) | Beta(1.8,1) |
| $\lambda_{\text{discrete}}$ | 0.1 | 0.1 |
| Learning rate | 0.001 | 0.001 |
| Learning rate schedule | N/A | N/A |
| Optimizer | AdamW | AdamW |
| Weight decay | 0.0 | 0.0 |
| EMA weight | 0.999 | 0.999 |
| Batch size | 32 | 32 |
| # Grad. accum. steps | 1 | 1 |
| # Rotation augs. | 8 | 8 |
| Effective batch size | 4,096 | 2,048 |
| # GPUs | 16 (80GB) | 8 (80GB) |
| Training duration | 168h | 168h |
| # Sampling steps | 100 | 100 |
| $g(t)$ | $\frac{1}{t+0.01}$ | $\frac{1}{t+0.01}$ |
| $\gamma_g$ | 0.01 | 0.01 |

Table 9: Finetuning hyperparameters across different downstream prediction tasks, optionally trained in sequential order.

| Hyperparam. | Mol. Properties (80M ❋ + 20M ◖) → | Mol. / Mat. Properties (80M ❋ + 20M ◖) → | Mol. / Mat. Energy & Forces (80M ❋ + 220M ◖) |
|---|---|---|---|
| # Dedicated params. | 18,104,595 | 18,104,595 | 217,255,140 |
| Aux. hidden size | 512 | 512 | 1024 |
| # Aux. transformer blocks | 4 | 4 | 8 |
| # Attn. heads | 8 | 8 | 8 |
| Train $t$ distribution | max(Beta(1.8,1), 0.98) | max(Beta(1.8,1), 0.98) | max(Beta(1.8,1), 1.0) |
| $\lambda_{\text{forces}}$ | N/A | N/A | 5.0 |
| Learning rate | 0.001 | 0.001 | 0.0003 |
| Learning rate schedule | N/A | N/A | N/A |
| Optimizer | AdamW | AdamW | AdamW |
| Weight decay | 0.0 | 0.0 | 0.0 |
| EMA weight | N/A | N/A | N/A |
| Batch size | 128 | 32 | 72 |
| # Grad. accum. steps | 2 | 4 | 1 |
| # Rotation augs. | 8 | 8 | 1 |
| Effective batch size | 4,096 | 2,048 | 1,152 |
| # GPUs | 2 (40GB) | 2 (80GB) | 16 (80GB) |
| Training duration | 48h | 192h | 120h |

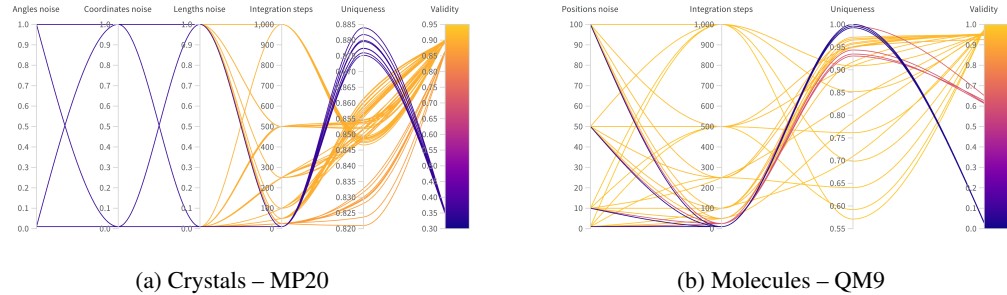

(a) Crystals – MP20

(b) Molecules – QM9

Figure 5: **Tuning inference hyperparameters for best performance.** Best generative modeling results for crystals and molecules, balancing sample validity and uniqueness, are generally achieved with a limited number of integration steps (e.g., $T = 100$), a high white noise scale $\gamma_g$ (e.g., 50) for continuous, non-periodic atom positions, and a low noise scale (e.g., 0.01) for continuous, periodic modalities such as fractional coordinates.

# D   EVALUATION METRICS

**Model development metrics for assessing crystal generation quality.**   To evaluate the quality of generated crystals during (iterative) model *development*, we adopt the methodology proposed by Xie et al. (2022), Miller et al. (2024), and Joshi et al. (2025). This process involves generating a sample of 1,000 crystals and then assessing them based on their validity and uniqueness. These metrics are described below:

- **Structural Validity**: This metric represents the percentage of crystals where all pairwise atomic distances are at least 0.5 and the crystal volume is 0.1 or greater.

- **Compositional Validity**: In accordance with the SMACT framework (Davies et al., 2019), this is the percentage of crystal compositions that successfully maintain both charge neutrality and electronegativity balance.

- **Overall Validity**: This is the percentage of crystals that satisfy both the structural and the compositional validity criteria mentioned above. Note that this metric is adopted in Figures 3 and 5.

- **Unique**: This measures the percentage of crystals that are structurally unique. Uniqueness is determined via an all-to-all comparison using the Structure Matcher tool within PyMatGen (Ong et al., 2013).

**Model evaluation metrics for assessing crystal generation quality.**   To evaluate the quality of generated crystals during (thorough) model *evaluation*, we adopt the methodology proposed by Duval et al. (2025). This process involves generating a sample of 2,500 crystals and then assessing them based on their validity, uniqueness, novelty, energy, and stability. These metrics are detailed below:

- **Charge Validity**: This metric ensures structures are charge-balanced using oxidation state analysis and bond valence calculations.

- **Distance Validity**: This validates that atomic distances exceed minimum thresholds based on atomic radii.

- **Coordination Validity**: This checks if coordination numbers match expected values for each element.

- **Physical Validity**: This validates density, lattice parameters, crystallographic format, and symmetry.

- **Overall Validity**: This represents the percentage of crystals meeting each of the evaluation validity criteria above.

- **Unique**: This measures the percentage of *valid* crystals that are also structurally unique. Uniqueness is determined via an all-to-all comparison using PyMatGen's Structure Matcher tool.

- **Novel**: This assesses the percentage of *valid*, *unique* crystals that are also structurally novel. Novelty is determined via an all-to-all comparison between the generated crystals and the LeMat-Bulk reference set (Siron et al., 2025) using the Structure Matcher tool of PyMatGen.

- **Formation Energy**: This denotes the average formation energy across multiple MLIPs, namely Orb-v3 (Rhodes et al., 2025), MACE-MP (Batatia et al., 2025), and UMA (Wood et al., 2025).

- **Mean Energy Above Hull**: This represents the average energy above hull across multiple MLIPs (i.e., Orb-v3, MACE-MP, UMA).

- **Relaxation Stability**: This signifies the root mean square deviation (RMSD) between original and relaxed atomic positions.

- **Stability Ratio**: This denotes the fraction of structures with energy above hull $\leq 0$ eV/atom (thermodynamically stable).

- **Stable, Unique, & Novel (SUN)**: This represents the fraction of structures that are simultaneously valid, stable (energy above hull $\leq 0$ eV/atom), unique, and novel, with the number of structures that meet each of these criteria shown in proceeding parentheses.

- **Metastability Ratio**: This signifies the fraction of structures with energy above hull $\leq 0.1$ eV/atom (metastable).

- **Metastable, Unique, & Novel (MSUN)**: This denotes the fraction of structures that are simultaneously valid, metastable (energy above hull $\leq 0.1$ eV/atom), unique, and novel, with the number of structures that meet each of these criteria shown in proceeding parentheses.

**Model development/evaluation metrics for assessing molecule generation quality.** For the evaluation of generated molecules, we utilize the protocol developed by Hoogeboom et al. (2022), Daigavane et al. (2024), and Joshi et al. (2025). From a sample of 10,000 molecules, we calculate rates for validity and uniqueness, and also determine success rates for seven distinct sanity checks provided by the PoseBusters toolkit (Buttenschoen et al., 2024):

- **Validity**: This measures the percentage of generated molecules for which a canonical SMILES representation can be successfully identified using RDKit (Landrum, 2013).

- **Uniqueness**: Among the molecules deemed valid, this metric calculates the percentage that possess a unique SMILES string.

- **All-atoms Connected**: This check quantifies the percentage of molecules in which every atom is part of a single connected component, meaning a continuous path of bonds links all atoms.

- **Reasonable Bond Angles/Lengths**: This is the fraction of molecules where all bond angles and lengths fall within a tolerance range, specifically between 0.75 and 1.25 times the standard values derived from distance geometry.

- **Aromatic Rings Flatness**: This metric counts the percentage of molecules where all atoms in 5- or 6-membered aromatic rings are positioned within 0.25Å of their nearest shared plane.

- **Double Bonds Flatness**: This is the percentage of molecules where the atoms of aliphatic carbon-carbon double bonds, along with their four neighboring atoms, lie within 0.25Å of the closest shared plane.

- **Reasonable Internal Energy**: This check determines the percentage of molecules whose calculated internal energy does not exceed 100 times the average energy observed in an ensemble of 50 conformations of the input molecule (Harris et al., 2023).

- **No Internal Steric Clash**: This metric represents the percentage of molecules where the distance between any pair of non-covalently bound atoms is greater than 0.8 times the lower bound established by distance geometry.

In summary, the validity and uniqueness metrics primarily confirm the chemical processability of the generated molecules by RDKit. In contrast, the PoseBusters sanity checks scrutinize the physical realism of the generated 3D structures, applying a range of criteria from geometric constraints to energetic plausibility.

## E  ADDITIONAL RESULTS

**Exploring metal-organic framework generation.** Table 10 investigates ZATOM-1's ability to generate metal-organic frameworks (MOFs), a novel class of materials with several promising applications (Kitagawa et al., 2014). With minimal hyperparameter tuning, we find that ZATOM-1 can generate valid MOFs despite their numerous chemical and geometric constraints, such as requiring a balance of metal, carbon, and hydrogen atoms without steric clashes. Nonetheless, the model (as currently evaluated) has not fully converged, indicating that its proportion of successfully generated MOFs could likely be improved with further training.

**Balancing multi-task molecule and material property prediction.** Table 11 presents the results of performing a second stage of finetuning for multi-task molecule and ($\Delta\varepsilon$) materials property prediction using QM9 and Matbench, starting from ZATOM-1's first stage of QM9 finetuning for multi-task molecule property prediction. In exchange for marginal molecule property prediction performance, these results demonstrate that ZATOM-1's generative *multi-task* property predictions can

Table 10: **Metal-organic framework generation results on QMOF.** We report sanity checks from MOFChecker (Jin et al., 2025) for 1,000 sampled MOFs, comparing methods trained only on QMOF for at least 10,000 epochs. (↑/↓ indicate higher/lower is better, respectively, and * denotes a model that has not fully converged)

| Test% | QMOF ADiT | QMOF ZATOM-1* |
|---|---|---|
| Has carbon ↑ | 100.0 | 100.0 |
| Has hydrogen ↑ | 99.6 | 100.0 |
| Has atomic overlap ↓ | 8.3 | 12.8 |
| Has overcoord. C ↓ | 23.6 | 3.2 |
| Has overcoord. N ↓ | 1.5 | 0.2 |
| Has overcoord. H ↓ | 1.0 | 3.1 |
| Has undercoord. C ↓ | 60.0 | 75.3 |
| Has undercoord. N ↓ | 39.1 | 31.5 |
| Has undercoord. rare earth ↓ | 0.4 | 0.0 |
| Has metal ↑ | 100.0 | 100.0 |
| Has lone molecule ↓ | 72.9 | 88.5 |
| Has high charge ↓ | 0.9 | 0.8 |
| Has suspicious terminal oxo ↓ | 2.6 | 1.9 |
| Has undercoord. alkali ↓ | 1.0 | 0.2 |
| Has geom. exposed metal ↓ | 7.0 | 2.8 |
| Validity rate (all passed) ↑ | 15.7 | 8.4 |

Table 11: **Property prediction for 3D molecules (QM9) / materials (Matbench) with joint molecule and materials finetuning.** Results are reported as *test* mean absolute errors (for ZATOM-1, with standard deviations over three runs with different random seeds). † denotes using different data partitions. / represents a task for which both (QM9) molecule and (Matbench) material property labels are available. The best (or second-best) results for each (optimized or unoptimized) category are in **bold** (or underlined). Notably, finetuning jointly for multi-task molecule and ($\Delta\varepsilon$) materials property prediction marginally trades off ZATOM-1's molecule property prediction performance for expanded applicability of its property predictions to both molecules and materials simultaneously.

| Model / Metrics ↓ | # Params | $\alpha$ ($a_0^3$) | $\Delta\varepsilon$ (meV) | $\varepsilon_{HOMO}$ (meV) | $\varepsilon_{LUMO}$ (meV) | $\mu$ (D) | $C_\nu$ (cal/mol K) | $G^{ATOM}$ (meV) | $H^{ATOM}$ (meV) | $R^2$ ($a_0^2$) | $U^{ATOM}$ (meV) | $U_0^{ATOM}$ (meV) | $ZPVE$ (meV) |
|---|---|---|---|---|---|---|---|---|---|---|---|---|---|
| **Optimized single-task learning** | | | | | | | | | | | | | |
| DimeNet++† | 5M | .044 | 33.0 / **199** | 25 | 20 | .030 | .023 | 8.00 | 7.00 | .331 | 6.00 | 6.00 | 1.21 |
| EGNN† | 1M | .071 | 48.0 / —— | 29 | 25 | .029 | .031 | 12.0 | 12.0 | .106 | 12.0 | 11.0 | 1.55 |
| PaiNN | 1M | .045 | 46.0 / —— | 28 | 20 | .012 | .024 | **7.35** | **5.98** | .066 | **5.83** | **5.85** | 1.28 |
| TorchMD-NET | 7M | .059 | 36.0 / —— | 20 | 18 | .011 | .026 | 7.62 | 6.16 | **.033** | 6.38 | 6.15 | 1.84 |
| SphereNet | 2M | .046 | 32.0 / —— | 23 | 18 | .026 | **.021** | 8.00 | 6.00 | .292 | 7.00 | 6.00 | **1.12** |
| SEGNN† | 1M | .060 | 42.0 / —— | 24 | 21 | .023 | .031 | 15.0 | 16.0 | .660 | 13.0 | 15.0 | 1.62 |
| EQGAT | - | .053 | 32.0 / —— | 20 | 16 | .011 | .024 | 23.0 | 24.0 | .382 | 25.0 | 25.0 | 2.00 |
| Equiformer | 4M | .046 | 30.0 / —— | 15 | 14 | .011 | .023 | 7.63 | 6.63 | .251 | 6.74 | 6.59 | 1.26 |
| EquiformerV2 | 11M | .050 | **29.0** / —— | **14** | **13** | .010 | .023 | 7.57 | 6.22 | .186 | 6.49 | 6.17 | 1.47 |
| PⱭNITA | - | **.038** | 30.4 / —— | 16 | 15 | .012 | .024 | 8.63 | 8.04 | .235 | 8.67 | 8.31 | 1.29 |
| Platonic Transformer | - | .049 | 37.4 / —— | 22 | 17 | **.010** | .024 | 12.0 | 12.0 | .222 | 11.9 | 13.0 | 1.30 |
| **Unoptimized single-task learning** | | | | | | | | | | | | | |
| AIM-STL† | - | .181 | —— / —— | 61 | 54 | **.067** | .072 | 66.2 | 63.9 | **.503** | 64.2 | 58.8 | **4.54** |
| **Unoptimized multi-task learning** | | | | | | | | | | | | | |
| AIM-MTL-Scalar† | - | .268 | —— / —— | 59 | 71 | .089 | .103 | 113 | 112 | 4.18 | 112 | 112 | 9.82 |
| AIM-MTL-Matrix† | - | .251 | —— / —— | 61 | 72 | .088 | .103 | 112 | 118 | 4.07 | 116 | 116 | 12.2 |
| QM9-pretrained ZATOM-1 (OURS) | 80M ❄ + 20M 🔥 | .105±.000 | 52.0±0.1 / 362±1 | 37±0.1 | 36±0.1 | .102±0.001 | .054±0.000 | 47.2±0.2 | 48.5±0.2 | 3.93±0.00 | 48.6±0.2 | 48.7±0.2 | 5.03±0.01 |
| Jointly pretrained ZATOM-1 (OURS) | 80M ❄ + 20M 🔥 | **.095±.000** | **46.2±0.1 / 354±1** | **34±0.1** | **32±0.1** | .087±.000 | **.047±.000** | **39.2±0.2** | **42.1±0.2** | 3.33±0.01 | **41.6±0.2** | **41.6±0.2** | 4.62±0.02 |

effectively be applied to both 3D molecules and materials, to our knowledge a first-of-its-kind result. In contrast, for materials property prediction in Table 4 of the main text, we report zero-shot $\Delta\varepsilon$ results for the model weights performing best specifically for multi-task molecule property prediction, to illustrate the broad applicability of ZATOM-1's pretrained molecule and material embeddings.

**Extension to energies and forces.** Table 12 explores ZATOM-1's ability to predict energies and forces to serve as an MLIP. Through this preliminary set of experiments, we find that the performance of ZATOM-1's material force predictions for MPtrj slightly surpasses that of Orb-v1, a state-of-the-art, pretrained model for this dataset. Results for OMol25 demonstrate that ZATOM-1 can effectively balance the performance of its force predictions for molecules and materials jointly, highlighting its potentially broad applicability in molecular and materials sciences. Despite these promising first steps, ZATOM-1's energy predictions could be notably improved, suggesting that its representations derived through generative pretraining on QM9 & MP20 are most informative of (equivariant) atomistic forces. As potential future directions, exploring bespoke additive noise schedules for each of the model's input modalities during finetuning, or pretraining using more structurally diverse datasets such as OMol25 and LeMat-Bulk, could potentially address this gap between energy and force finetuning performance for both molecules and materials.

Table 12: **Energy and force prediction for 3D materials (MPtrj) / molecules (OMol25-4M).** The results are reported as *validation* Energy (in meV or meV/Atom) and Force (in meV/Å) mean absolute error (MAE). All models directly predict forces, rather than using the gradient of their predicted energies with respect to input 3D coordinates for conservative force prediction, to considerably increase their computational efficiency.

| Model | # Params | Energy MAE/Atom (meV) ↓ | Energy MAE (meV) ↓ | Force MAE (meV/Å) ↓ |
|---|---|---|---|---|
| **Material Interatomic Potential Prediction on MPtrj** | | | | |
| Non-pretrained Orb-v1 | ∼30M | - | 350.00 | 122.00 |
| Denoising-pretrained Orb-v1 | ∼30M | - | 210.00 | 32.00 |
| Jointly pretrained ZATOM-1 | 80M �֍ + 220M ◖ | 112.81 | 2631.22 | 30.92 |
| **Molecule Interatomic Potential Prediction on OMol25-4M** | | | | |
| eSEN-md-d. | ∼51M | 1.32 | 77.13 | 6.78 |
| Jointly pretrained ZATOM-1 | 80M ✖ + 220M ◖ | 34.45 | 2512.78 | 62.99 |

## F VISUALIZATION

**Generated crystals, molecules, and MOFs.** Figures 6 and 7 show representative crystals and small molecules, respectively, sampled from ZATOM-1 trained jointly on the QM9 and MP20 datasets. Additionally, Figure 8 visualizes MOFs generated by ZATOM-1 trained on the QMOF dataset. Overall, ZATOM-1 generates compositionally diverse crystals across multiple spacegroups and chemically valid, structurally diverse molecules, indicating that it learns a rich generative latent space rather than a redundant set of representations.

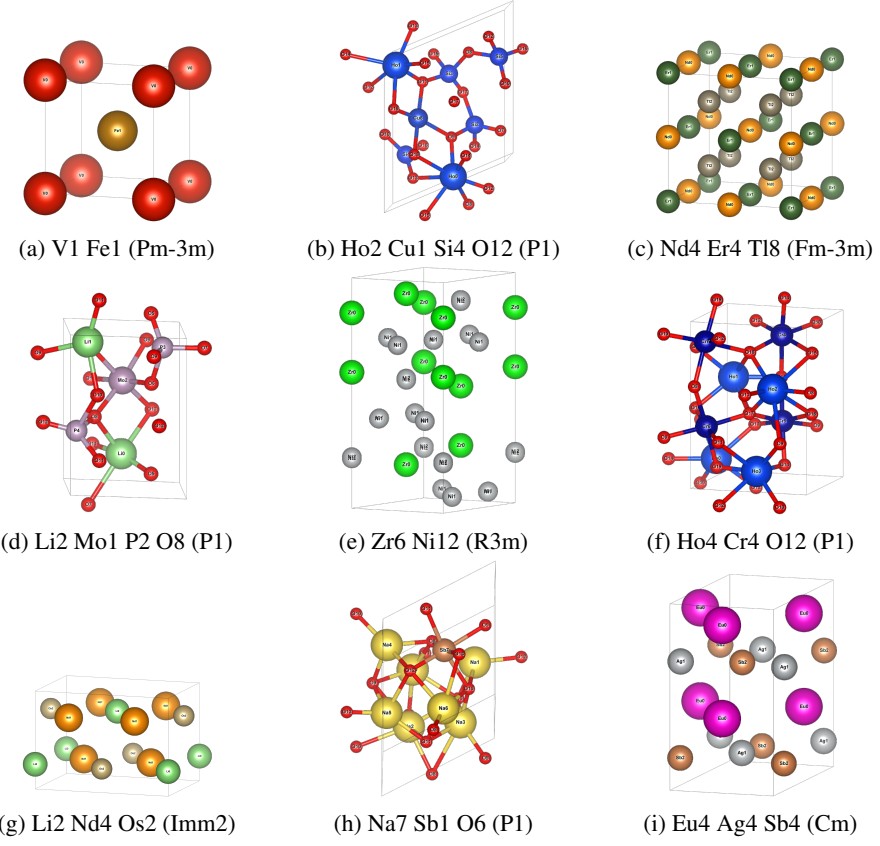

(a) V1 Fe1 (Pm-3m)  (b) Ho2 Cu1 Si4 O12 (P1)  (c) Nd4 Er4 Tl8 (Fm-3m)

(d) Li2 Mo1 P2 O8 (P1)  (e) Zr6 Ni12 (R3m)  (f) Ho4 Cr4 O12 (P1)

(g) Li2 Nd4 Os2 (Imm2)  (h) Na7 Sb1 O6 (P1)  (i) Eu4 Ag4 Sb4 (Cm)

Figure 6: Generated crystals from ZATOM-1 trained jointly on MP20 materials and QM9 molecules.

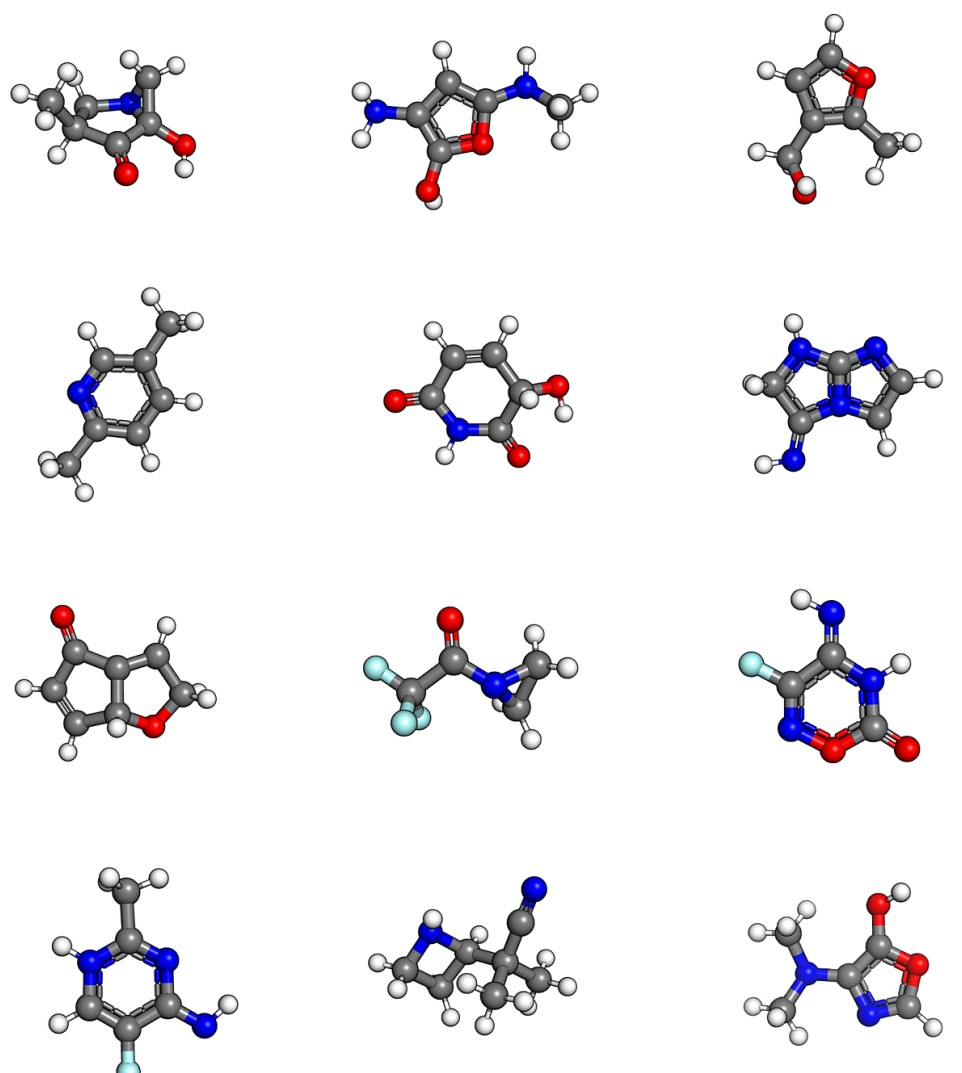

Figure 7: Generated molecules from ZATOM-1 trained jointly on MP20 materials and QM9 molecules.

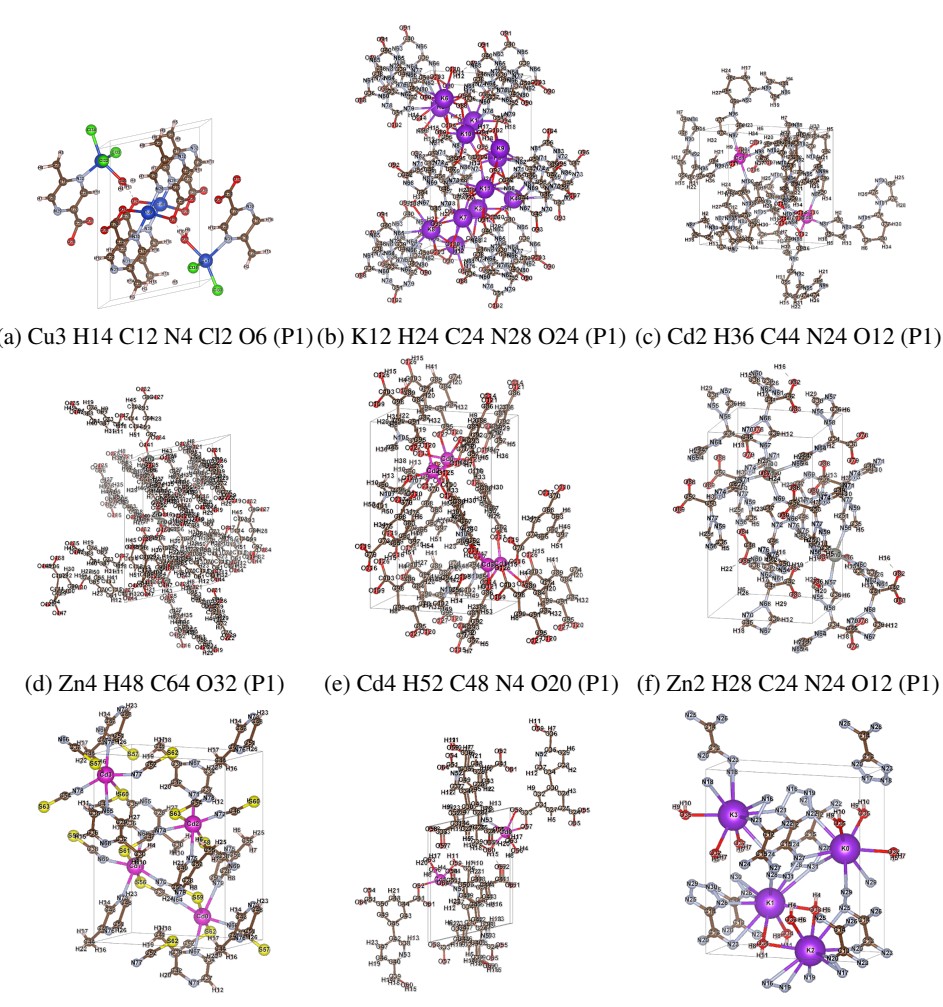

(a) Cu3 H14 C12 N4 Cl2 O6 (P1)  (b) K12 H24 C24 N28 O24 (P1)  (c) Cd2 H36 C44 N24 O12 (P1)

(d) Zn4 H48 C64 O32 (P1)  (e) Cd4 H52 C48 N4 O20 (P1)  (f) Zn2 H28 C24 N24 O12 (P1)

(g) Cd4 H24 C28 S8 N16 (P1)  (h) Cd2 H22 C28 N2 O12 (P1)  (i) K4 H8 C4 N16 O4 (P1)

Figure 8: Generated metal-organic frameworks from ZATOM-1 trained on QMOF metal-organic frameworks.

