# OpenReview forum: "Zatom-1: A Multimodal Flow Foundation Model for 3D Molecules and Materials"
_ICLR.cc/2026/Workshop/FM4Science — ICLR 2026 Workshop FM4Science Poster_

### Official Review · Reviewer_SRXL · 2026-02-19
**An interesting work of using transformer models to unify the generative and predictive modeling for Molecules and Materials**

**Rating:** 7
**Confidence:** 3

**Review:**

This paper introduces ZATOM-1, which is the first transformer-based foundation model that is designed to unify the generative and predictive learning of 3D molecules and materials. They designed a novel model that can jointly handle discrete atom types and continuous 3D geometries for both periodic and non-periodic systems, providing flexibility for material and molecule representations.  They used the transformer architecture and trained it with a multimodal flow matching objective. The complete training procedure contains the multimodal flow pretraining and predictive finetuning. Their results have found that ZATOM-1 matches or outperforms specialized baselines in both generative and predictive tasks while achieving a reduction in generative inference time by more than an order of magnitude. It is so far the first foundation model capable of learning across the distinct domains of periodic materials and non-periodic 3D molecules, making it very valuable for the scientific community. I would recommend acceptance.

---

### Official Review · Reviewer_sSTj · 2026-02-21
**Review on ZATOM-1: A Multimodal Flow Foundation Model for 3D Molecules and Materials**

**Rating:** 6
**Confidence:** 4

**Review:**

# Quality
The paper is technically sound overall, with a well-structured multimodal flow matching formulation. The experimental evaluation is extensive, covering generation (MP20, QM9, GEOM-Drugs, QMOF), property prediction (QM9, Matbench), and energy/force prediction (OMol25, MPtrj).

# Clarity

The paper is generally well-written and well-organized which has a logical progression from problem formulation to architecture to experiments. The two-algorithm pseudocode presentation (Algorithms 1 and 2) is commendable for reproducibility.

## Minor Issue

- Appendix B.2, Table 6b: "jontly trained" is a typo for "jointly trained."

## Missing References

- **UniGenX** (Zhang, Li, Luo et al., 2025, "UniGenX: a unified generative foundation model that couples sequence, structure and function to accelerate scientific design across proteins, molecules and materials"): a concurrent unified generative foundation model spanning molecules and materials with property prediction. This paper must be discussed to properly scope the "first" claim.

# Originality

The key novelty is the combination of component ideas: (1) multimodal flow matching across molecules and materials, (2) using generative pretraining as initialization for predictive tasks, and (3) doing this within a single unified Transformer architecture. These components come from prior works.

# Significance

This work addresses an important problem of unifying chemical modeling across molecules and materials and is a promising proof-of-concept for the unified generative-predictive paradigm in 3D ambient space.


# Strengths
- The central contribution is novel. The two-stage pipeline (Figure 1) is a clean design that naturally enables transfer.
- Table 4 provides strong evidence that joint molecule+material pretraining improves molecular property prediction.
- Figure 2 shows ZATOM-1 generates crystals/molecules very fast. This changes the practical utility of generative sampling.
- Table 2b indicates the model has learned physically realistic 3D geometries, not just valid SMILES graphs.
- Figure 3 shows near-perfect Pearson/Spearman correlations across three model sizes, suggesting the architecture will benefit from further scaling.

# Weaknesses
- The "first" and "foundation model" claims require more careful qualification considering the UniGenX. The paper should acknowledge and distinguish from the UniGenX.
- The energy prediction results (Table 12) is weak.
- Crystal generation novelty rates are low in Table 1.

---

### Decision · Program_Chairs · 2026-03-03

Accept (Poster)